# Systematic genetic characterization of the human PKR kinase domain highlights its functional malleability to escape a poxvirus substrate mimic

**Michael James Chambers[1,2], Sophia B Scobell[1], Meru J Sadhu[1]***

[1]Center for Genomics and Data Science Research, National Human Genome Research Institute, National Institutes of Health, Bethesda, United States; [2]Department of Microbiology & Immunology, Georgetown University, Washington, United States

## eLife Assessment

This **important** revised report describes the control of the activity of the RNA-activated protein kinase, PKR, by the Vaccinia virus K3 protein. A strength of the manuscript is the powerful combination of a classic yeast-based assay with high-throughput sequencing and its **convincing** experimental use to characterize large numbers of PKR variants, now with improved controls for potential biases. A minor current limitation that the authors may address in the future is the scope of the screen in terms of the segments of PKR included.

*For correspondence:
meru.sadhu@nih.gov

**Competing interest:** The authors declare that no competing interests exist.

**Abstract** Evolutionary arms races can arise at the contact surfaces between host and viral proteins, producing dynamic spaces in which genetic variants are continually pursued. However, the sampling of genetic variation must be balanced with the need to maintain protein function. A striking case is given by protein kinase R (PKR), a member of the mammalian innate immune system. PKR detects viral replication within the host cell and halts protein synthesis to prevent viral replication by phosphorylating eIF2α, a component of the translation initiation machinery. PKR is targeted by many viral antagonists, including poxvirus pseudosubstrate antagonists that mimic the natural substrate, eIF2α, and inhibit PKR activity. Remarkably, PKR has several rapidly evolving residues at this interface, suggesting it is engaging in an evolutionary arms race, despite the surface's critical role in phosphorylating eIF2α. To systematically explore the evolutionary opportunities available at this dynamic interface, we generated and characterized a library of 426 SNP-accessible nonsynonymous variants of human PKR for their ability to escape inhibition by the model pseudosubstrate inhibitor K3, encoded by the vaccinia virus gene *K3L*. We identified key sites in the PKR kinase domain that harbor K3-resistant variants, as well as critical sites where variation leads to loss of function. We find K3-resistant variants are readily available throughout the interface and are enriched at sites under positive selection. Moreover, variants beneficial against K3 were also beneficial against an enhanced variant of K3, indicating resilience to viral adaptation. Overall, we find that the eIF2α-binding surface of PKR is highly malleable, potentiating its evolutionary ability to combat viral inhibition.

## Introduction

Molecular interfaces between interacting host and pathogen proteins are atomic battlegrounds that give rise to evolutionary arms races (***Daugherty and Malik, 2012***). Successful infection typically involves

**eLife digest** When viruses replicate, mutations can be introduced into their genetic information. These mutations can confer viruses with an advantage during infection: the proteins a virus uses to stick to host cells may change to bind better, allowing the virus to introduce itself into cells more easily; or the virus may lose a specific pattern in its surface, making it harder for the immune system to detect. These changes can make a virus more contagious or more deadly and can lead to the immune system acquiring mutations to counteract the viruses' own.

In humans, a protein called PKR alerts host cells if a virus is present by adding a chemical tag to a second protein called eIF2α, which can halt the production of proteins in the cell, thus stopping viral replication. A type of virus, known as the poxvirus, has evolved a way to stop this 'alarm' by producing a protein called K3, which mimics eIF2α to intercept PKR and prevent the inhibition of protein synthesis. Both PKR and K3 seem to be evolving rapidly over time, gaining mutations that provide a competitive edge over one another. This is particularly interesting because PKR can adopt genetic changes that evade K3, but it can still recognize is natural substrate, eIF2α, which is not evolving.

The building blocks or amino acid residues in PKR that are rapidly evolving have been characterized. Additionally, several different versions or variants of PKR have been identified, where a change in a single residue allows K3 evasion, whilst maintaining its ability to halt protein synthesis.

Chambers et al. wanted to know how easily PKR can mutate to evade K3. The researchers used yeast to see how well variants of PKR functioned, since halting protein production impairs yeast growth. They screened 426 different versions of PKR, each with a change in a single building block, to see whether they retained the ability to target eIF2α, including when pitted against two variants of K3.

The results showed that PKR is extremely genetically pliable: it can change its surface to keep recognizing eIF2α while evading K3 in many different ways. Chambers et al. also characterized variants of PKR that were non-functional (because they could not bind eIF2α) and distinguished them from those that were susceptible to K3. They found that variants of PKR that could evade K3 could also evade an improved version of the protein. This suggests that the PKR variants that evade K3 are resilient.

Understanding this pattern of resiliency – both to mutations in PKR and to variants of K3 –may in the future aid therapeutic designs.

---

critical interactions between host and pathogen proteins, such as binding to cell-surface receptors or inhibition of cellular defense mechanisms. In such scenarios, the host benefits from genetic mutations that disrupt the interaction while the pathogen benefits from mutations that maintain the interaction. The pursuit of genetic mutations by the host and pathogen can develop into an evolutionary arms race where the accumulation of mutations leads to abnormally high sequence diversity, which is used to identify sites experiencing positive selection.

A key member of the antiviral innate immune repertoire in vertebrates is protein kinase R (PKR) (*Metz and Esteban, 1972*; *Clemens, 1997*; *Balachandran et al., 2000*). PKR contains two N-terminal double-stranded RNA (dsRNA) binding domains and a C-terminal kinase domain, separated by a long linker (*Dar et al., 2005*). The presence of dsRNA in the cytoplasm is a hallmark of viral infection (*Weber et al., 2006*). PKR binds dsRNA, initiating PKR homodimerization and autophosphorylation, which activates the kinase (*Meurs et al., 1990*; *Kaufman, 2000*; *Dey et al., 2005*; *Lemaire et al., 2008*). Active PKR binds and phosphorylates eIF2α on Ser51 (*Figure 1A*), a member of the translation initiation machinery. Phosphorylation of eIF2α leads to a halt in protein synthesis, thereby preventing viral replication (*Kaufman, 2000*; *Hershey, 1991*; *Sudhakar et al., 2000*).

Vertebrate PKR homologs have many sites under positive selection (*Rothenburg et al., 2009*), including residues at its binding surface with eIF2α (*Figure 1B*). This is remarkable given the critical nature of this interface and that eIF2α is an unvarying protein under purifying selection (*Elde et al., 2009*). Many viruses encode PKR inhibitors to evade its antiviral activity, some of which are known to interact with PKR at its eIF2α-binding interface (*Langland et al., 2006*; *Nallagatla et al., 2011*; *Hull and Bevilacqua, 2016*). One example is vaccinia virus, whose well-characterized pseudosubstrate inhibitor protein K3 is encoded by the *K3L* gene (*Figure 1C*; *Beattie et al., 1991*; *Carroll et al.,*

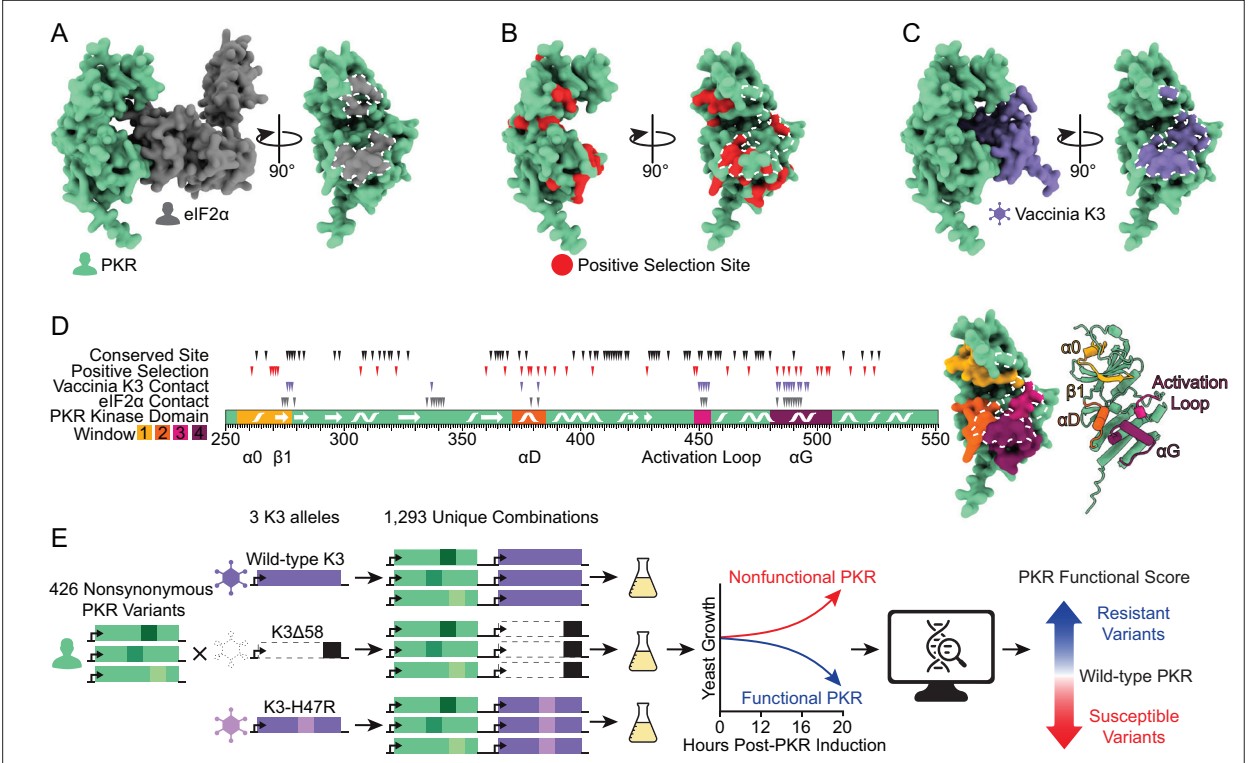

**Figure 1.** Exploring genetic variants of human PKR against the pseudosubstrate antagonist vaccinia K3. (**A**) (Left) AlphaFold2 model of the PKR kinase domain (green) bound to its target, eIF2α (gray). (Right) Rotated 90°, the eIF2α binding surface of PKR is shaded gray and outlined in white dashed line. (**B**) PKR kinase domain with vertebrate positive selection sites highlighted in red (*Rothenburg et al., 2009*), with the K3 binding surface delineated with white dashed line. (**C**) AlphaFold2 model of the PKR kinase domain bound to vaccinia K3 (purple). Rotated 90°, the K3 binding surface of PKR is shaded purple and outlined in white dashed line. (**D**) (Left) The PKR kinase domain with four windows highlighted in which we designed 426 nonsynonymous variants. Secondary structures occurring in these windows (alpha helices α0, αD, and αG, beta strand β1, and the kinase activation loop) are marked (*Dar et al., 2005*). The positions of conserved sites and sites under positive selection (*Rothenburg et al., 2009*) are denoted above with black and red triangles, respectively, and the positions of sites within 5 angstroms of either K3 or eIF2α in an Alphafold2 binding model are denoted with purple and gray triangles, respectively. Note that the modeled interaction between PKR and eIF2α resembles the pre-phosphorylation state in which the phosphoacceptor loop of eIF2α has not entered the PKR active site, and thus there are likely additional sites in PKR that contact eIF2α. (Right) Mutated windows are highlighted on the surface of the PKR kinase domain, with the K3 binding surface delineated with a white dashed line, and in cartoon form with secondary structures marked. (**E**) Methodological approach to explore the effects of variants of human PKR in the presence of different K3 alleles. 426 nonsynonymous variants of PKR were generated and paired with wild-type K3, K3Δ58, and K3-H47R. Variant effects were characterized using a high-throughput yeast growth assay and massively parallel sequencing.

The online version of this article includes the following figure supplement(s) for figure 1:

**Figure supplement 1.** Composition of the PKR variant library.

**Figure supplement 2.** Calculation of PKR functional scores from yeast growth assay.

**Figure supplement 3.** Replication of PKR variants paired with K3 alleles.

**Figure supplement 4.** Systematic generation of PKR variants using mixed-base primer tile sets.

**Figure supplement 5.** Multiple unique barcode sequences were attached to each PKR variant.

**Figure supplement 6.** Assembly of PKR variant library using variant tile sets and barcode primers.

**Figure supplement 7.** Alphafold2 multimer predictions used to identify PKR sites proximal to eIF2α and K3.

*1993*; *Davies et al., 1993*). Vaccinia K3 mimics the structure and sequence of eIF2α to inhibit PKR and prevent the halt of protein synthesis (*Carroll et al., 1993*; *Sharp et al., 1997*; *Dar and Sicheri, 2002*; *Kawagishi-Kobayashi et al., 1997*). Thus, despite evolutionary pressure on PKR to recognize the conserved eIF2α substrate, variation at the same interface would be evolutionarily beneficial if it allowed evasion of viral pseudosubstrate antagonists. Indeed, previous studies have identified some single-residue variants at sites under positive selection in the PKR kinase domain that maintain recognition of eIF2α yet resist vaccinia K3 (*Rothenburg et al., 2009*; *Elde et al., 2009*; *Seo et al., 2008*).

However, without comprehensively exploring the genetic landscape at this interface, it is unclear how readily PKR can mutate to evade viral antagonists while preserving its kinase function.

Here we use a prospective, systematic, and unbiased approach to map SNP-accessible PKR variants in the kinase domain that are resistant or susceptible to the vaccinia wild-type K3, nonfunctional K3, and the enhanced mutant allele K3-H47R. We found many novel K3-resistant variants, especially at sites under positive selection, and note a strong correlation between variants that resist wild-type K3 and the enhanced allele. We also identified sites in the PKR kinase domain where variants increased susceptibility to vaccinia K3. Essential PKR residues that did not tolerate variation were infrequent despite the critical nature of this interface; they were primarily found surrounding the ATP-binding site. These results paint a portrait of PKR balancing the tension between conserved function and evasion of viral inhibitors through mutational resilience in its substrate interface.

## Results

### Generation of nonsynonymous PKR variant library

We systematically made genetic variants in four windows of interest in the kinase domain of PKR (*Figure 1D*). The four windows were selected based on having sites within five angstroms of eIF2α or K3 and sites under positive selection across vertebrate PKR homologs as identified by *Rothenburg et al., 2009*. Our windows of interest include helix αD and αG, as well as the glycine-rich loop and activation loop surrounding the ATP-binding site. In these windows, we focused on testing 426 variants that could be acquired with a single nucleotide change from the predominant human PKR allele (Genbank M85294.1; *Figure 1—figure supplement 1*), as those are the most likely to be evolutionarily sampled.

Overexpression of PKR in the budding yeast *Saccharomyces cerevisiae* is toxic, as PKR auto-activates and shuts down protein synthesis by phosphorylating yeast eIF2α, which is highly conserved between yeast and humans (*Chong et al., 1992*; *Dever et al., 1993*). This toxicity can be partially counteracted by co-expression of vaccinia K3 (*Kawagishi-Kobayashi et al., 1997*). Thus, yeast provides a useful experimental system to test the functionality of PKR variants: variants that maintain the ability to phosphorylate eIF2α while escaping K3 inhibition will further arrest growth, while variants that lose eIF2α kinase activity or become more susceptible to K3 will improve growth.

We sought to leverage this assay in a high-throughput manner to determine the functionality of PKR variants by tracking their abundances in a pool over time. We generated the 426 single-residue variants of PKR on a single-copy plasmid with a galactose-inducible promoter in a plasmid pool (*Figure 1E*). Each unique plasmid in the pool carried a DNA barcode randomly generated during cloning. We used long-read circular consensus sequencing to pair DNA barcodes with PKR variants, identifying an average of 43 barcodes per variant (*Figure 1—figure supplement 1*). We then created three libraries by combining the PKR variant library with wild-type K3, nonfunctional K3, or the enhanced mutant allele K3-H47R (*Kawagishi-Kobayashi et al., 1997*; *Elde et al., 2012*). The K3 alleles were expressed from a constitutive *TDH3* promoter. We transformed the three PKR-K3 paired libraries in duplicate into yeast grown in glucose media, such that PKR expression was repressed. We then inoculated the libraries in liquid galactose media to induce PKR expression, and sampled cells at 0, 12, 16, and 20 hr post-inoculation. We quantified the abundance of the PKR variants at each timepoint using high-throughput short-read sequencing of the DNA barcodes. For each PKR variant we calculated a functional score against each K3: PKR variants that evade K3 inhibition and reduce yeast growth have high functional scores, while PKR variants that are non-functional or are inhibited by vaccinia K3, and thus are associated with good yeast growth, have low functional scores (*Figure 1—figure supplement 2*). The correlation of functional scores between replicates 1 and 2 was very strong (Pearson correlation coefficient >0.98 for each K3 allele, *Figure 1—figure supplement 3*), so we combined the reads from replicates 1 and 2 to generate PKR functional scores for subsequent analyses.

### Many PKR variants evade K3 while maintaining kinase function

We first examined how each PKR variant performed against wild-type K3. We were encouraged to find that variants previously characterized as resistant to K3-H47R (*Seo et al., 2008*) were resistant to wild-type K3 in our screen (*Figure 2A*). We found many additional K3-resistant variants across all four tested windows and validated select PKR variants using serial dilution to measure yeast growth

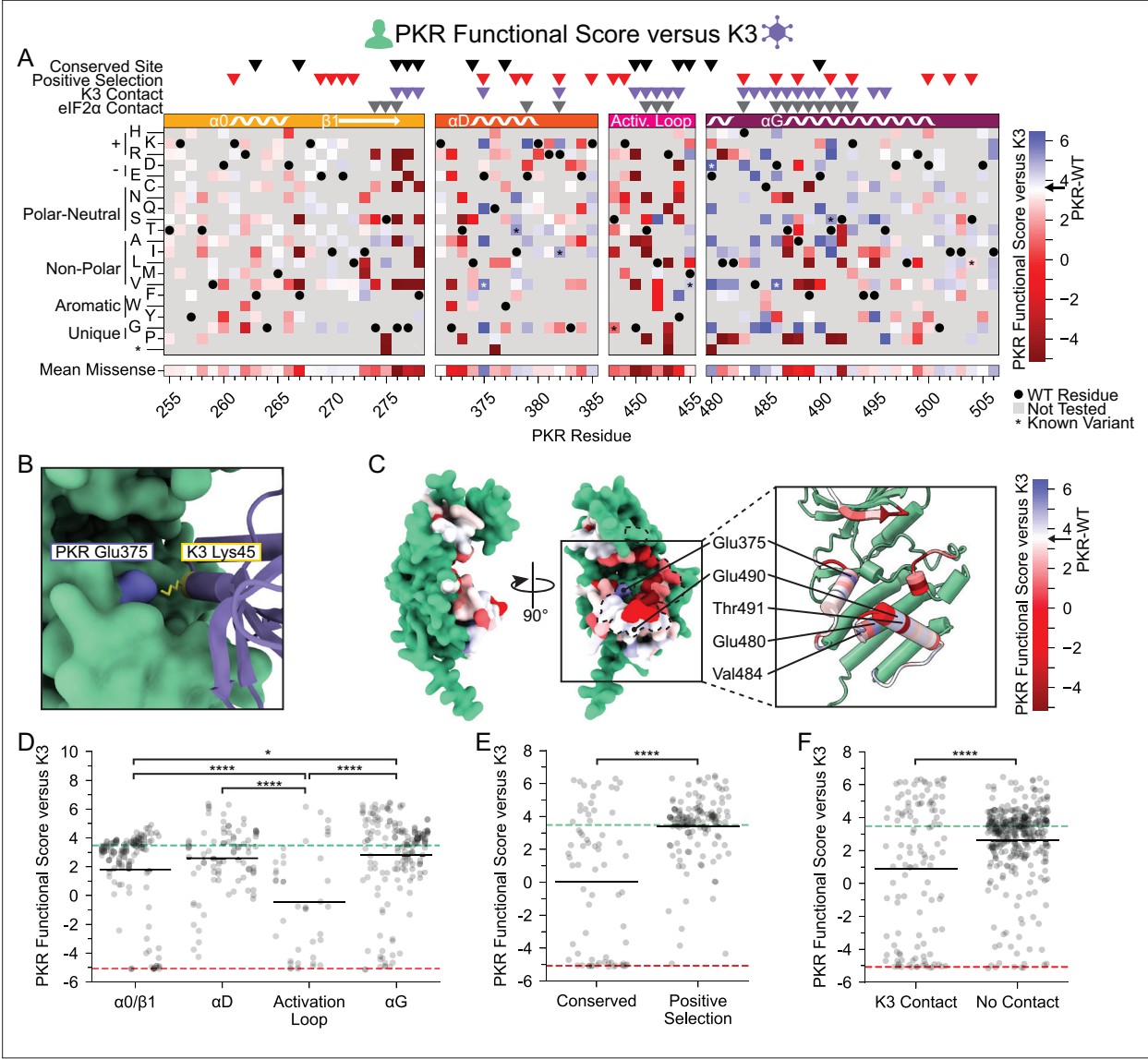

**Figure 2.** PKR variants that evade K3 and maintain kinase function are enriched at positive selection sites and helices αD and αG. (**A, C**) PKR functional scores versus K3 are colored ranging from susceptible (red) to WT-like (white) to resistant (blue). (**A**) Heatmap of PKR variants with cells colored by the PKR functional score versus K3 for SNP-accessible variants. Wild-type PKR residues and untested variants are denoted with black circles and gray squares, respectively. Previously characterized K3-H47R-resistant variants are noted with an asterisk (*Seo et al., 2008*). (**B**) Attractive electrostatic interaction between PKR-Glu375 (blue) and K3-Lys45 (yellow) in the AlphaFold2 model of PKR bound to K3. (**C**) Surface structure of the PKR kinase domain with sites colored by the mean PKR functional score versus K3 for missense variants. The K3 binding surface is delineated with black dashed line. (Inset) Location of K3-resistant sites cited in the text. (**D–F**) Strip plots of PKR functional score versus K3 for variants, with green and red dashed lines representing mean scores for WT PKR and nonsense variants, respectively. Variants are partitioned by nearest secondary structural element (**D**), level of conservation in vertebrates (**E**), or predicted contact with K3 (**F**). Points in D are ordered left to right by their position along the kinase domain, whereas they are randomly jittered along the x-axis in E and F. * $p<0.05$, **** $p<0.0001$, Tukey's HSD (**D**) and two-sample t-test (**E, F**).

The online version of this article includes the following figure supplement(s) for figure 2:

**Figure supplement 1.** Experimental validation of the K3 resistance phenotypes of PKR variants.

**Figure supplement 2.** Bimodal distribution of PKR functional scores at K3 contact sites.

(*Figure 2—figure supplement 1*). Notably, every variant made at Glu375 increased evasion of K3, with the exception of Glu375Asp. In the AlphaFold2-predicted complex, Glu375 forms an attractive electrostatic interaction with Lys45 of K3 (*Figure 2B*). Consistent with our findings, this predicted interaction would be disrupted by all tested variants at site 375 aside from the negatively charged aspartic acid. Interestingly, Glu375 is the only site under positive selection in vertebrates that was

proximal to K3 but not eIF2α in the AlphaFold2 models, making it a prime candidate for disrupting K3 inhibition without affecting eIF2α phosphorylation.

We found a large number of K3-resistant variants in the vicinity of PKR's helix αG (*Figure 2C and D*). Compared to other protein kinase structures, eIF2α kinases have a noncanonical helix αG that is extended by one full turn and rotated 40° counterclockwise with a 5 angstrom translation relative to its C-terminus, owing to a reduced αF-αG linker (*Dar et al., 2005*). The noncanonical orientation of helix αG contributes to both eIF2α binding specificity and Ser51 presentation to PKR (*Dar et al., 2005*). It has been proposed that mutations causing movement in helix αG might be especially capable of disrupting K3 binding due to the rigid structure of K3 between its contact points on helix αG and the PKR active site. In contrast, PKR mutations that move helix αG could be tolerated in the context of eIF2α binding because eIF2α contacts PKR's active site using a flexible loop (*Dar et al., 2005*). We observe a gradient of PKR functional scores along the helix αG region, starting with highly resistant or susceptible variants (positions 480–492) and shifting to neutral wild-type-like scores (493-500) (*Figure 2A*). We identified four sites within or near the N-terminus of helix αG for which many variants evade K3: Glu480, Val484, Glu490, and Thr491 (*Figure 2C*).

We looked at what features of PKR were associated with sites of K3-resistant variants and were excited to find that sites under positive selection in vertebrates were strongly enriched for K3-resistant variants relative to sites conserved across vertebrate PKRs (*Figure 2E*). Comparing K3 contact and non-contact sites, we find significantly lower functional scores at K3 contact sites (*Figure 2F*). However, functional scores at K3 contact sites form a bimodal distribution (Hartigan's dip test statistic

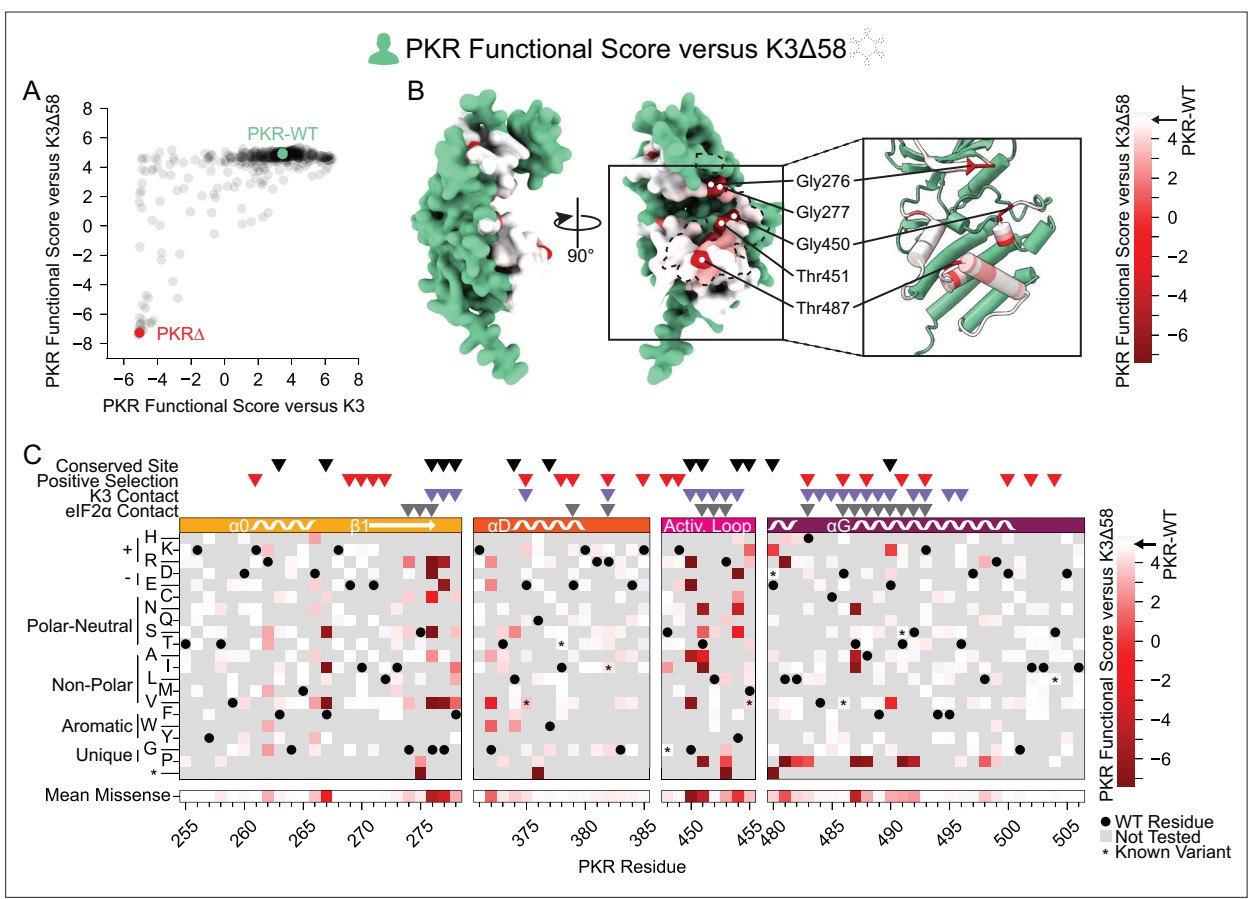

**Figure 3.** Few nonfunctional PKR variants identified in the absence of K3 inhibition. (**A**) Scatter plot showing each variant's PKR functional score versus wild-type K3 plotted against its PKR functional score versus K3Δ58. The data point for PKR-WT is colored green and a data point representing the average of the four nonsense variants (PKRΔ) is colored red. (**B, C**) PKR functional scores versus K3Δ58 are colored ranging from nonfunctional (red) to WT-like (white). (**B**) Surface structure of the PKR kinase domain with sites colored by the mean PKR functional score versus K3Δ58 for missense variants. The K3 binding surface is delineated with a black dashed line. (Inset) Location of highly constrained sites cited in the text. (**C**) Heatmap of PKR variants with cells colored by the PKR functional score versus K3Δ58 for each variant.

= 0.105, p<0.0001). Considering only functional variants (PKR functional score > –2) we find that variants at K3 contact sites have significantly higher functional scores than those at non-contact sites (*Figure 2—figure supplement 2*). Thus, variants at K3 contact sites tend towards large effects either improving or diminishing PKR function versus K3.

## Sampled sites in the PKR kinase domain are largely resilient to genetic variation

To understand the constraint placed on the PKR kinase domain, we examined variants that failed to maintain PKR function. We screened our PKR variant library against a nonfunctional K3 (hereafter referred to as K3Δ58) in which the first 173 bases were deleted, removing the first 58 residues, including the initial methionine. When paired with K3Δ58, most PKR variants were comparable to wild-type PKR (*Figure 3A*), suggesting the tested regions were not highly functionally constrained for kinase activity despite their proximity to eIF2α during its phosphorylation. The few highly constrained sites include Gly276 and Gly277, which compose the glycine-rich loop above the ATP-binding site; Gly450 and Thr451 in the activation loop; and Thr487 in helix αG, which stabilizes helix αG (*Dar et al., 2005*) and induces the conformational change in eIF2α required for its phosphorylation (*Figure 3B*; *Dey et al., 2011*). We also found proline variants to generally disrupt PKR function (*Figure 3C*). We found nonfunctional variants of varying degrees in all four windows of interest, but with very few in

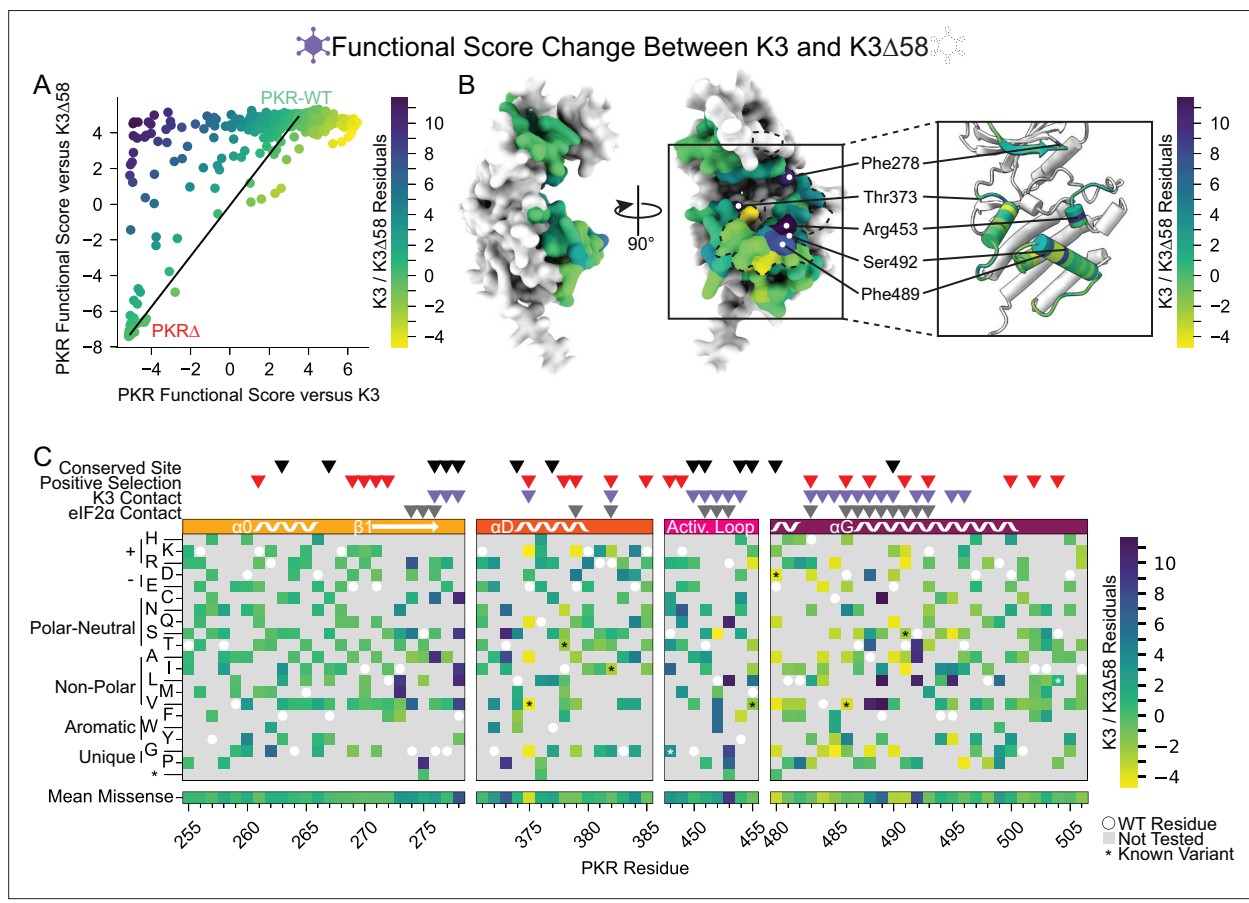

**Figure 4.** Identification of PKR sites highly susceptible to K3 inhibition. (**A**) A line was drawn connecting the data point for PKR-WT to the position of the average data point for the four nonsense variants (PKRΔ) from the data in the scatter plot from *Figure 3A*, with all other data points colored by their residual from that line (K3 /K3Δ58 Residuals), ranging from K3-susceptible (purple) to K3-indifferent (green) to K3-resistant (yellow). This color scheme is used in panels B and C. (**B**) Surface structure of the PKR kinase domain with sites colored by the mean K3 /K3Δ58 residuals for missense variants. The K3 contact site is delineated with a black dashed line. (Inset) Location of K3-susceptible sites cited in the text. (**C**) Heatmap of PKR variants with cells colored by the K3 /K3Δ58 residuals for each variant.

The online version of this article includes the following figure supplement(s) for figure 4:

**Figure supplement 1.** Sequence similarity between human eIF2α and vaccinia K3.

helix αD and the C-terminus of helix αG (*Figure 3C*), consistent with these regions having pliable secondary structures in which variants can evade K3 while not disrupting eIF2α recognition.

More PKR variants had low functional scores in the presence of functional K3 (*Figure 2A*) than in its absence (*Figure 3C*). We plotted the functional scores against each other (*Figure 4A*), and found that variants with low PKR functional scores versus K3 separated into two classes: some switched to high functional scores in the absence of K3, indicating they caused enhanced susceptibility to K3 but otherwise retained PKR function, while others had low functional scores in both conditions, indicating they had diminished eIF2α kinase activity irrespective of K3 antagonism. We quantified this effect by calculating residuals from the line connecting PKR nonsense variants to PKR-WT, which highlights variants whose functional scores diverged between the K3 conditions. We find that the activation loop and the start of helix αG, which were less tolerant of mutation when paired with wild-type K3 (*Figure 2D*), contain a mix of variants that lose function outright or that increase susceptibility to K3 (*Figures 3B–C and 4B–C*). Variants at five sites were especially susceptible to K3 inhibition: Phe278, Thr373, Arg453, Phe489, and Ser492 (*Figure 4B*). None appear under positive selection across vertebrates, and Phe278 is conserved across vertebrate PKR homologs. Interestingly, Phe278 is predicted to form a pi interaction with His47 of K3, which aligns to Arg52 of eIF2α (*Figure 4—figure supplement 1*; *Kawagishi-Kobayashi et al., 1997*). Mutation of K3-H47 to arginine, matching eIF2α, is known to improve K3's antagonism of PKR (*Kawagishi-Kobayashi et al., 1997*; *Elde et al., 2012*), suggesting PKR can discriminate against K3 based on its difference from eIF2α at this position. This is consistent with our observation of the importance of Phe278 specifically in the presence of K3.

## K3-resistant variants were largely resistant to enhanced K3-H47R

We also characterized PKR variants in the presence of an enhanced mutant allele of K3, K3-H47R (*Figure 5*). This mutant allele was first identified in a genetic screen of K3 mutants as an enhanced antagonist of PKR (*Kawagishi-Kobayashi et al., 1997*) and independently identified in a directed evolution screen (*Elde et al., 2012*). We sought to understand the resilience of improved PKR variants to alternative K3 alleles by examining the concordance between beneficial PKR variants against wild-type K3 and K3-H47R. We found strong correlation between PKR functional scores versus K3 and K3-H47R (Spearman correlation = 0.832, *Figure 5A*). Indeed, many of the K3-resistant variants are also resistant to K3-H47R (*Figure 5B–D*), suggesting resilience of PKR to allelic variation in K3. A general change is the struggle for PKR to overcome the enhanced inhibition of K3-H47R, compressing the functional score range between wild-type-like and nonfunctional scores. This nonlinear relationship between the functional scores in the two K3 conditions (*Figure 5—figure supplement 1*) led to some differences in the patterns of resistance to the two K3 alleles. For instance, positively selected PKR sites did not have significantly higher PKR functional scores versus K3-H47R than conserved sites (*Figure 5E*), a sharp contrast to the PKR functional scores versus wild-type K3 (*Figure 2E*). This result appears to be caused by the many wild-type-like scores at positively selected sites now being similar to the many nonsense-like scores at conserved sites. We also noted that variants at sites proximal to K3 had slightly higher functional scores than those at non-contact sites (*Figure 5F*), the opposite of what we observed against wild-type K3 (*Figure 2F*). This could result from wild-type-like PKR variants being enriched at non-contact sites. Overall, the strong correlation of functional scores versus K3 and K3-H47R suggests K3-resistant variants would often be resilient against alternate K3 alleles.

Previous work identified twelve PKR variants with enhanced resistance to K3-H47R (*Seo et al., 2008*), nine of which were included in our experiment (*Figure 5—figure supplement 2*). We examined the other nonsynonymous variants introduced at these sites and found that most sites contained additional novel resistant variants, indicating the previously uncovered individual mutants reflect sites of general opportunity for pseudosubstrate evasion.

## Discussion

Here, we characterize the local evolutionary space available to the human innate immunity protein PKR to maintain functionality while evading the well-studied vaccinia virus antagonist, K3. One strategy viruses leverage against PKR is pseudosubstrate inhibition, in which proteins like K3 bind PKR by structurally mimicking its natural substrate eIF2α, inhibiting PKR kinase activity. Interestingly, this binding interface contains many residues under positive selection. However, positively selected genetic

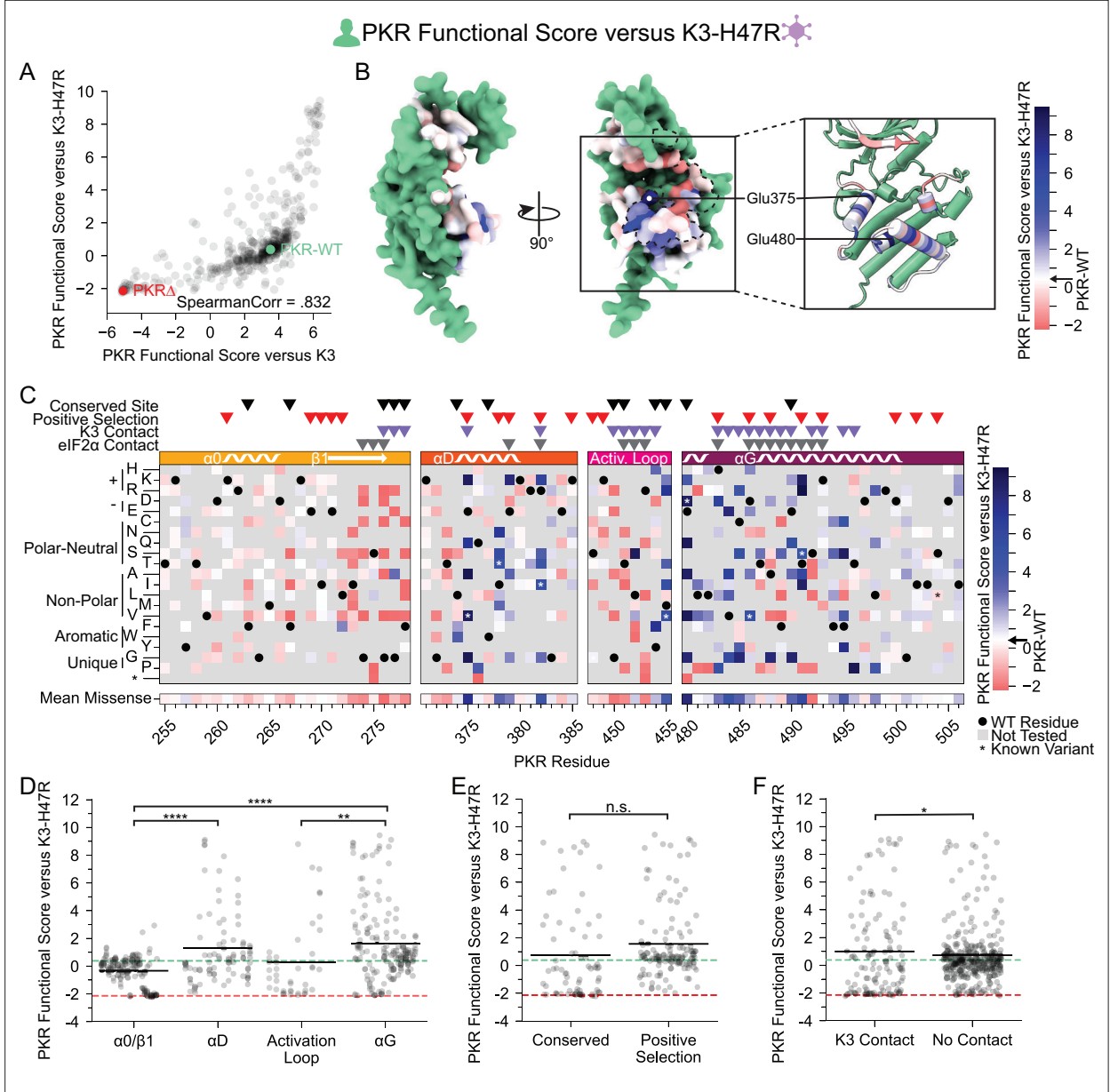

**Figure 5.** PKR variants that are K3-resistant are also largely K3-H47R-resistant. (**A**) Scatter plot of PKR functional scores versus wild-type K3 plotted against PKR functional scores versus K3-H47R. The data point for PKR-WT is colored green and a datapoint representing the average of the four nonsense variants (PKRΔ) is colored red. (**B, C**) PKR functional score versus K3-H47R ranging from susceptible (red) to WT-like (white) to resistant (blue). (**B**) Surface structure of the PKR kinase domain with sites colored by the mean PKR functional score versus K3-H47R for missense variants. The K3 binding surface is delineated with a black dashed line. (Inset) Location of K3-H47R-resistant sites cited in the text. (**C**) Heatmap of PKR variants with cells colored by the PKR functional score versus K3-H47R for each variant. (**D–F**) Strip plots of PKR functional scores versus K3-H47R for variants, as plotted in *Figure 2D–F*. Variants are partitioned by nearest secondary structural element (**D**), level of conservation in vertebrates (**E**), or predicted contact with K3 (**F**). Points in D are ordered left to right by their position along the kinase domain, whereas they are randomly jittered along the x-axis in E and F. * p<0.05, ** p<0.01, **** p<0.0001, Tukey's HSD (**D**) and two-sample t-test (**E, F**).

The online version of this article includes the following figure supplement(s) for figure 5:

**Figure supplement 1.** Differing patterns of resistance between wild-type K3 and K3-H47R.

**Figure supplement 2.** Additional variants at the sites of previously identified K3-H47R-resistant variants often also conferred resistance to K3.

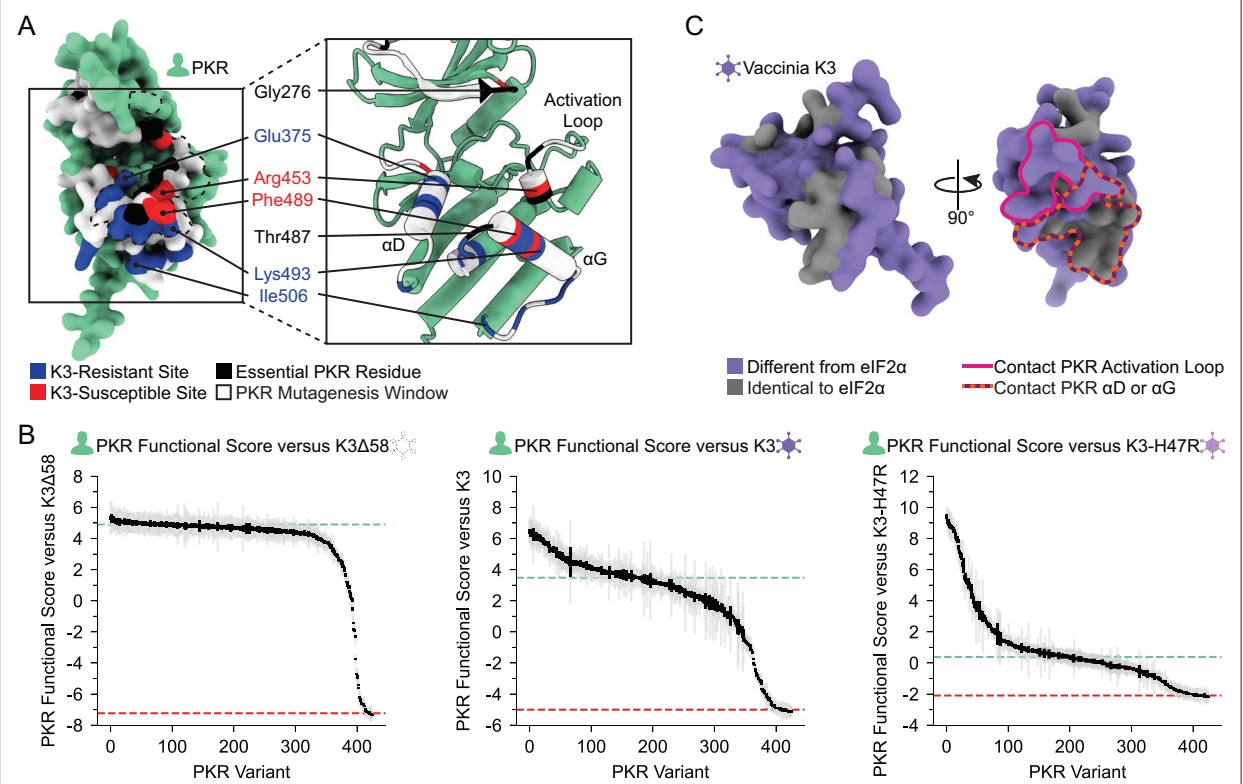

**Figure 6.** Unified spatial view of highlighted PKR sites across the K3-binding interface. (**A**) Positions of sites classified as essential residues (black), K3-resistant (blue), and K3-susceptible (red) based upon the above analyses. All other tested sites are marked white. (Left) The vaccinia K3 binding surface of PKR is outlined with a black dashed line. (**B**) PKR functional scores versus K3Δ58 (Left), K3 (Center), and K3-H47R (Right), with each vertical line representing a PKR variant. Variants are sorted by functional score from high (left) to low (right). Gray and black lines indicate the standard deviation and standard error, respectively, in PKR functional scores across barcodes associated with the given variant. (**C**) (Left) AlphaFold2 model of Vaccinia K3 with surface residues colored as different from eIF2α (purple) or identical to eIF2α (gray) based on sequence alignment (*Figure 4—figure supplement 1*). (Right) Rotated 90°, K3 residues that contact the PKR activation loop or helix αD or αG are outlined in pink and dashed orange and burgundy, respectively.

variants across the binding interface would need to maintain binding with PKR's natural substrate while evading viral pseudosubstrate antagonists. In this study, we quantified the abundance of such 'tightrope-walking' genetic variants relative to the abundance of functionally deleterious variants, which highlighted particular sites of opportunity and constraint within the kinase domain.

We identified many sites containing variants that evade vaccinia K3 without losing eIF2α targeting. Resistant variants clustered around helices αD and αG, secondary structures with sites under positive selection and where resistant variants have been previously characterized (*Rothenburg et al., 2009*; *Seo et al., 2008*). Overall, we found few genetic variants that impair the kinase function of PKR (*Figure 3*), underscoring the genetic resilience of PKR, with functionally constrained sites mostly localized to the vicinity of the ATP-binding site. Finally, we found a strong correlation between PKR variants that are resistant against both K3 and the enhanced antagonist K3-H47R, suggesting the K3-resistant variants are resilient to K3 allelic variation.

We identified sites across the PKR kinase domain that were K3-resistant, K3-susceptible, or essential to eIF2α phosphorylation (*Figure 6A*). Sites enriched with genetic variants that improve evasion of K3 were classified as K3-resistant sites, such as Glu375 and Glu480. We find clusters of these sites located across helices αD and αG, at the binding surface of K3 and eIF2α. Positive selection sites are enriched for K3-resistant variants. There were sites under positive selection where we did not observe much K3 resistance, such as sites 269–272, which may suggest this region is antagonized by a completely different viral antagonist. Still, we found these sites to be generally able to adopt variants that maintain PKR function (*Figure 3*), making them prime 'tightrope-walking' sites for thwarting any alternate viral inhibitors that interact at those sites. We also identified some K3-resistant sites that are

not under positive selection, such as Glu480, which is in fact conserved across vertebrate PKR homologs and the four human eIF2α kinases (*Dey et al., 2005*; *Rothenburg et al., 2009*). Perhaps it has an important role in a function not tested in our assay, such as avoiding off-target phosphorylation of other human proteins.

Our analysis does not account for variation in PKR expression that may result from the introduced variation. Previous work demonstrated that K3-resistant PKR variants were not enhanced due to increased expression (*Rothenburg et al., 2009*). We note that we did not observe much variation in PKR functional scores in the absence of K3 inhibition (*Figure 3A*), which suggests PKR variants did not often substantially modulate inherent properties of PKR, like its expression.

We found PKR variants that were not tolerated, although perhaps fewer than we would expect given the kinase domain is responsible for carrying out its critical function. These came in two flavors: variants that increased susceptibility to K3, and variants that were deleterious for PKR function. We were surprised to find sites enriched for K3-susceptible variants, such as Phe278, Arg453, and Ser492. These could be sites where the wild-type residues help discriminate between eIF2α and K3. These sites do not cluster together; however, when projected onto the surface of PKR they appear to be intermingled with essential residues (*Figure 6A*). This could suggest that while they are not needed to recognize eIF2α in normal conditions, they are well-positioned to discriminate between eIF2α and a pseudosubstrate inhibitor. Finally, the correlation of PKR functional scores between K3 and K3-H47R suggests resistant variants are resilient. This agrees with prior work screening a few PKR variants against vaccinia K3-H47R and the K3 homolog from variola virus (*Rothenburg et al., 2009*). We would be curious to know if this pattern holds for more highly diverged K3 orthologs (*Haller et al., 2014*) and other pseudosubstrate inhibitors, such as Ranavirus vIF2α (*Rothenburg et al., 2011*).

The genetic pliability observed in the PKR kinase domain is reminiscent of that seen with another innate immunity factor, TRIM5α, which is locked in an evolutionary arms race with retroviral capsids (*Sawyer et al., 2005*; *Tenthorey et al., 2020*). Both PKR and TRIM5α have access to 'rolling hills' of resistant variants (*Figure 6B*), in opposition to sharp cliffs that are highly optimized (*Tenthorey et al., 2020*). Notably, unlike PKR, TRIM5α does not have to balance pathogen conflict with enzymatic function. We also note the structural context in which we identified K3-resistant sites, with many being found within secondary structures (e.g. Glu375, Il378, and Glu379 in helix αD, and Glu490, Lys493, and Thr491 in helix αG), whereas in TRIM5α, as well as the immune proteins MxA and NLRP1 (*Colón-Thillet et al., 2019*; *Chavarría-Smith et al., 2016*), positively selected sites and improved variants have been described in structurally flexible loops and linkers. We also considered contrasts between PKR and ACE2, a host enzyme with sites under positive selection likely due to being targeted by coronaviruses for cell entry (*Demogines et al., 2012*; *Damas et al., 2020*; *Frank et al., 2022*). However, the ACE2 sites under positive selection are not located near the catalytic domain of the enzyme, in contrast to PKR (*Lan et al., 2020*; *Chan et al., 2020*; *Heinzelman and Romero, 2021*). Ultimately, it appears that the resolution of this conundrum is that PKR maintains pliability in this region despite its criticality. This pliability is likely made possible by the flexibility of eIF2α's Ser51 loop in which the phosphorylation site resides; indeed, the loop is unstructured in the co-crystal of PKR and eIF2α (*Dar et al., 2005*). As such, while we still don't have a clear picture of what phosphorylation of eIF2α looks like, we can appreciate the flexible nature of eIF2α during its interaction with PKR (*Dar et al., 2005*; *Elde et al., 2009*; *Seo et al., 2008*; *Dey et al., 2011*). Further biochemical and structural investigation will be necessary to interrogate the ability of variants in PKR to disrupt interactions with K3 while maintaining eIF2α targeting. We note that while we were able to identify many sites of interest across the PKR kinase domain, variation at many other sites in PKR remains to be explored and may harbor allosteric effects on pseudosubstrate resistance.

Barring possible variation in under-studied populations, there is little allelic diversity of PKR in the human population, with all nonsynonymous variation listed as rare in gnomAD (*Chen et al., 2024*). PKR does exhibit sequence diversity in comparisons across species, including primates (*Elde et al., 2009*; *Jacquet et al., 2022*). We expect viruses could exert selective pressure on rare heterozygous variation in PKR, as we would expect beneficial variants in PKR that evade pseudosubstrate inhibitors to be dominant, given that a small amount of eIF2α phosphorylation is capable of halting protein synthesis (*Siekierka et al., 1984*).

We found an enrichment for vaccinia K3-resistant variants at sites previously found to be under positive selection (*Rothenburg et al., 2009*; *Elde et al., 2009*). In effect, these are two independent

approaches that both pinpoint these sites as genetically pliable: positive selection analysis retro-spectively considers existing variation in nature across multiple homologs, whereas deep mutational scanning prospectively tests novel variants in a single homolog. We and others (*Tenthorey et al., 2020*; *Colón-Thillet et al., 2019*) view deep mutational scanning experiments as a complementary approach to positive selection analysis. One strength of deep mutational scanning is its potential to illuminate sites of opportunity against pathogens that have not exerted historical evolutionary pressure. It can also allow for the testing of insertions or deletions (*Ogden et al., 2019*) which are generally discarded in positive selection analysis. On the other hand, a limitation of the deep muta-tional scanning approach is that it only tests variant function in a limited set of conditions, when the evolutionary selection could be a symphonic signal made up of a myriad of unknown pressures. Deep mutational scanning also requires the function of the protein to be testable in a selectable assay. While proteins with cell-extrinsic or whole-organism phenotypes would be harder to test, recent advances in high-throughput genetic experimentation may allow deep mutational scans of those proteins as well (*Parvez et al., 2021*; *Popp et al., 2024*).

## Ideas and speculation

One might naively expect a PKR pseudosubstrate inhibitor to be a closer mimic of the natural PKR substrate, eIF2α, which would make it more difficult for PKR to distinguish it from eIF2α. One possi-bility is that K3 pseudosubstrate divergence from eIF2α has been selected to allow the inhibitor to bind PKR with even higher affinity than eIF2α does. But mutations that improve binding to one species' PKR homolog might decrease binding to the divergent PKR homologs from other species (*Park et al., 2019*; *Park et al., 2021*; *Peng et al., 2016*). Thus, pseudosubstrate inhibitors of gener-alist poxviruses may need to evolutionarily balance improving PKR binding for one host against main-taining PKR binding of multiple hosts. Interestingly, the structural component most diverged between K3 and eIF2α, a rigid helix in K3 in place of eIF2α's flexible phospho-acceptor site, primarily interacts with the residues around the activation loop of PKR (*Figure 6C*), which we found to be highly func-tionally constrained. In contrast, amino acids 72–83 of K3 are highly identical to eIF2α and interact primarily with PKR's divergent helix αG (*Figure 4—figure supplement 1*). Thus, vaccinia K3 may well have been evolutionarily constrained to avoid limiting host range. While the host range of vaccinia is not fully understood (*Smith, 2007*), current evidence suggests vaccinia is a generalist (*Haller et al., 2014*; *Park et al., 2021*). Poxviruses as a whole infect a very wide range of species, which suggests the ancestral K3 needed to be compatible with a wide host range.

Despite its pliability, PKR helix αG is believed to play a critical role in eIF2α binding and phosphor-ylation, as it induces a conformational shift in eIF2α that moves the Ser51 phospho-acceptor site into PKR's ATP-binding site. Based on our findings and positive selection analyses, this conformational shift is apparently amenable to variation in the PKR kinase domain. Perhaps the resilience of the binding surface between PKR and eIF2α could be informative for the budding field of de novo protein design, as resilient design is a critical consideration for vaccines and therapeutics faced with rapidly evolving infectious agents.

## Materials and methods

### PKR and K3 plasmid construction

We generated the plasmid MSp508_MCS as the base plasmid for all experiments by inserting a multiple-cloning site (MCS) into the single-copy plasmid YCp50, a gift from Mark Rose (*Rose et al., 1987*). The MCS contained seven unique restriction enzyme cut sites: AgeI, BsiWI, BstEII, NotI, MluI, PvuII, and SacI. YCp50 was digested using restriction enzymes EcoRI-HF (NEB Cat#R3101S) and BspDI (NEB Cat#R0557S) with rCutSmart buffer, followed by purification and size selection on a 1% agarose gel with 0.6 µg/mL ethidium bromide (BioRad Cat#1610433), selecting for the 7,961 bp band. The extracted band was purified using a QIAquick Gel Extraction kit (Qiagen Cat#28706) and eluted with 30 µL of water. For Gibson Assembly the purified band was combined with a single-stranded DNA oligonucleotide containing the MSC and homology arms spanning the digest site, named Oligo 1 (*Supplementary file 1*), in a 1:5 molar ratio in a total volume of 5 µL, along with an additional 5 µL 2 x Gibson Assembly Master Mix (NEB Cat#E2611L). The reaction mixture was incubated at 50 °C for 1 hr. One microliter of the Gibson Assembly reaction was transformed into 10 µL 5-alpha competent

*E. coli* (NEB Cat#C2987I) and plated onto 10 cm LB-AMP plates (IPM Scientific Cat#11006–016) and incubated overnight at 37 °C. Colonies were selected for overnight outgrowth in 3 mL LB-AMP (IPM Scientific Cat#11006–004), followed by a QIAprep spin miniprep (Qiagen Cat#27106) and Sanger sequencing to validate the plasmid sequence using the primer Oligo 2. This produced the plasmid MSp508_MCS.

We cloned wild-type PKR, wild-type K3, and K3-H47R into the MCS of MSp508_MCS to produce the MSp509_PKR-WT, MSp510_K3-WT, and MSp511_K3-H47R plasmids, respectively. The plasmid p1419 (*Kawagishi-Kobayashi et al., 1997*) contains the *PKR* gene (standard name *EIF2AK2*) under the control of the galactose-inducible pGAL10/CYC1 promoter for yeast expression. We PCR-amplified the PKR expression construct using primers (Oligos 3 and 4) to place pGAL10/CYC1-*PKR* on MSp508_MCS between AgeI and BstEII. Oligo 4 incorporated a synthetic terminator for *PKR*, Tsynth1 (*Curran et al., 2015*), a 28 bp spacer sequence, and a 26 bp barcode (CGCTTAATATGCAATGAAAT TGCTTA); the primers added 20 bp homology arms for cloning onto the MSp508_MCS plasmid. PCR was performed using the polymerase PfuUltraII (Agilent Technologies Cat#600674) for 30 cycles with an annealing temperature of 55 °C, followed by gel purification. MSp508_MCS was digested with restriction enzymes AgeI-HF (NEB Cat#R3552S) and BstEII-HF (NEB Cat#R3162S), followed by gel purification. The digested MSp508_MCS and pGAL10/CYC1-PKR fragments were joined via Gibson Assembly in a 1:5 molar ratio, followed by transformation into 5-alpha competent *E. coli* and plating onto LB-AMP. A resultant clone, validated by Sanger sequencing, was designated MSp509_PKR-WT.

To produce MSp510_K3-WT, the *K3L* coding sequence was amplified from the plasmid pC140 (*Kawagishi-Kobayashi et al., 1997*) using primers Oligo 5 and Oligo 6. Similarly, to produce MSp511_ K3-H47R, the *K3L-H47R* coding sequence was amplified from the plasmid pC407 using primers Oligo 6 and Oligo 7. pC140 and pC407 were gifts from Thomas Dever. We note that the *K3L* alleles on these plasmids carry a Val2Leu mutation relative to the VACV-WR *K3L* (Genbank AAO89313.1), which was introduced during the cloning of *K3L* on the original plasmid pTM1 (*Carroll et al., 1993*). Oligos 5 and 7 incorporated a synthetic terminator ($T_{synth8}$) for the K3 allele (*Curran et al., 2015*), a 28 bp nucleotide spacer sequence, a 26 bp barcode (ATCGTAATAGGTTTCCGGCTTGTTCG and ACAGGAAA TGCTTTCGGGGTTGTATT, respectively), and a 20 bp homology arm to MSp508_MCS. The pTDH3 promoter, also known as pGPD, was amplified from genomic DNA of the yeast strain BY4742 (*Brachmann et al., 1998*) using primers Oligo 8 and Oligo 9. K3-WT, K3-H47R, and pTDH3 were amplified using PfuUltraII, followed by gel purification. MSp508_MCS was digested with restriction enzymes NotI-HF (NEB Cat#R3189S) and PvuII-HF (NEB Cat#R3151S). Each K3 allele (K3-WT and K3-H47R) was joined with a pTDH3 promoter onto the digested MSp508_MCS via Gibson Assembly in a 1:5:5 molar ratio, followed by transformation into 5-alpha competent *E. coli* and plating onto LB-AMP. Resultant clones, validated by Sanger sequencing, were designated as MSp510_K3-WT and MSp511_K3-H47R.

MSp512_K3Δ58 was made by removing nucleotides 1–173 of the *K3L* sequence of YCp50_K3, leaving codons 59–88 to generate a nonfunctional *K3L* allele (denoted K3Δ58) under the pTDH3 promoter. The K3Δ58 sequence was amplified from the MSp510_K3-WT plasmid using primers Oligo 10 and Oligo 11. MSp510_K3-WT was digested with restriction enzymes NotI-HF and MluI-HF (NEB Cat#R3198S), followed by gel purification and size selection for the 8684 bp band. The K3Δ58 allele was joined with the MSp508_MCS digested fragment via Gibson Assembly in a 1:5 molar ratio, followed by transformation into 5-alpha competent *E. coli* and plating onto LB-AMP. A resultant clone was validated by Sanger sequencing and designated as MSp512_K3Δ58.

## Generation of the *PKR* variant library

We made the *PKR* variant library by 15 Gibson Assembly reactions using pairs of PCR amplicons targeting 4 windows of PKR protein sequence: 255–278 (5 pairs of reactions), 371–385 (3 pairs of reactions), 448–455 (2 pairs of reactions), and 480–506 (5 pairs of reactions). In each pair, one of the PCR reactions, PCR-1, utilized a pool of forward primers, termed a variant primer tile set, containing single-base deviations from the *PKR* sequence to generate all SNP-accessible variants in a stretch of X to Y consecutive codons (*Figure 1—figure supplement 4*). Each forward primer in a tile set was composed of a 20 bp homology arm, the variant region, and a priming region with a melting temperature of 50 °C. The reverse primer of all PCR-1 reactions had a constant 20 bp homology arm, a barcode region with 20 fully mixed bases, and a constant PCR-priming region homologous to the end of *PKR* with a melting temperature of 55 °C (*Figure 1—figure supplement 5*). Mixed bases in the

barcodes were limited to 5-nucleotide stretches to prevent the unintended generation of restriction sites, spacing these stretches with 'AA' or 'TT' sequences (*Liu et al., 2019*). The forward PCR-1 primer tile sets are listed in the *Supplementary file 1* as Oligos 12–26, and the reverse PCR-1 barcode primer is listed as Oligo 27. The paired PCR-2 reaction was an inverse PCR that generated the remainder of the *PKR* gene and plasmid backbone, with overlaps to the homology arms of PCR-1; the forward primer for PCR-2 reactions was Oligo 28, and the reverse primers were Oligos 29–43. Thus, Gibson Assembly of these PCR products would generate sets of *PKR* variants in defined segments of *PKR* along with random barcodes at the end of the gene (*Figure 1—figure supplement 6*).

We used custom Python (v3.8.18) scripts to design each variant-introducing primer, utilizing IUPAC degenerate base symbols (e.g. 'D' represents a mix of 'A', 'G', and 'T'; *Cornish-Bowden, 1985*; *Guido and Drake, 2009*). The introduced variant amino acids were all within a single base change from the canonical human *PKR* coding sequence (Genbank M85294.1:31–1686); we note the *PKR* sequence encoded on MSp509_PKR-WT differs somewhat from the canonical sequence through synonymous substitutions. Primers were synthesized by Integrated DNA Technologies, then were pooled manually into variant primer tiles sets.

PCR-1 reactions were amplified from the MSp509_PKR-WT plasmid using PfuUltraII polymerase with an annealing temperature of 45 °C. Each reaction generated a pool of *PKR* gene fragments containing a single nonsynonymous variant paired with a unique barcode sequence. In parallel, PCR-2 reactions amplified vector fragments from MSp509_PKR-WT plasmid using Herculase II polymerase (Agilent Technologies Cat#600677) using cycling conditions for vector targets >10 kb. PCR-1 and PCR-2 amplicons were digested with DpnI (NEB Cat#R0176S) for 2 hr at 37 °C to remove the MSp509_PKR-WT template DNA, followed by gel purification. Each PCR-1 amplicon was paired with its corresponding PCR-2 amplicon for Gibson Assembly in a 1:5 molar ratio.

Two µL of each Gibson Assembly was transformed into 50 µL of 5-alpha competent *E. coli* and all cells were plated on LB-AMP and incubated overnight at 37 °C. Colonies were counted on each plate and plates were bottlenecked at approximately 30 colonies per selected nonsynonymous variant (i.e. each nonsynonymous variant would be linked to approximately 30 unique barcode sequences). Plates were washed with 15 mL LB-AMP, cell densities were measured as the optical density at 600 nm ($OD_{600}$), and an equal number of cells (approximately $5 \times 10^9$ cells) from each reaction were pooled to form a single *PKR* variant library. A 200 mL LB-AMP outgrowth was grown to an $OD_{600}$ measurement of approximately 3, followed by a Qiagen MAXI plasmid prep (QIAGEN Cat#12963).

## PacBio sequencing of barcoded variant libraries

The *PKR* variant library was sequenced using the PacBio Sequel II instrument to identify the barcodes that were linked to each nonsynonymous variant. Sixteen micrograms of the *PKR* variant library were digested with restriction enzymes AgeI-HF and NotI-HF to create linear fragments of approximately 2100 bp for sequencing. The digest was incubated for 2 hr at 37 °C then purified using a QiaQuick PCR purification kit (Qiagen Cat#28106). Fragments of 2100 bp were size-selected using a SageELF instrument (Sage Science) before PacBio circular consensus sequencing (CCS) HiFi sequencing on a Sequel II instrument (Pacific Biosciences). From the CCS reads we generated a table of *PKR* barcodes paired with genetic variants using alignparse v0.2.6 and custom Python scripts (*Crawford and Bloom, 2019*).

## Combining *PKR* variant library with K3 alleles

We next cloned the three K3 alleles (K3, K3-H47R, and K3Δ58) onto the PKR variant library plasmids. The PKR variant library was digested with NotI-HF and PvuII-HF to generate a 10,270 bp receiver DNA fragment, while the K3 alleles were separately digested with BstEII-HF and SacI-HF (NEB Cat#R3156S) to generate a 1130 bp insert DNA fragment with 20 bp homology arms with homology to the receiver fragment for Gibson Assembly. For each of the reactions, 10 µg of plasmid DNA was digested with 5 µL of each restriction enzyme in a 50 µL reaction to generate a linear fragment, followed by gel purification. The PKR and K3 fragments were combined in three separate Gibson Assembly reactions in a 1:5 molar ratio with 100 ng of the PKR vector fragment in a reaction volume of 20 µL and incubated at 50 °C for 1 hr. A standard ethanol precipitation was performed on each reaction, adding 100 µL of 100% ETOH and incubating overnight at –20 °C before resuspending the pellets in 2 µL of water (*Sambrook and Russell, 2006*).

We transformed each of the three concentrated reactions into *E. cloni* 10 G SUPREME electro-competent cells (Lucigen Cat#60080–2) using a MicroPulser Electroporator (BioRad), with 2 µL of the concentrated reaction combined with 25 µL of competent cells. Following the electroporation and recovery, cells were plated onto two 15 cm LB-AMP plates. A 1:1000 dilution was plated with 100 µL of water on a 10 cm plate to estimate the number of transformed colonies across the two 15 cm plates, which produced an estimate of approximately 2 million colonies per electroporation. Plates were washed with 25 mL LB-AMP, combining the two plates for each of the reactions. Each library was expanded in 200 mL LB-AMP to an $OD_{600}$ measurement of approximately 3, followed by a Qiagen MAXI plasmid prep.

## Screening the PKR-K3 variant pairs in a yeast growth assay

All yeast growth was at 30 °C in CSM-Ura (Sunrise Science Products Cat#1004–010), and liquid cultures were shaken at 200 revolutions per minute. We transformed each of the three paired libraries (PKR +K3, PKR +K3Δ58, and PKR +K3 H47R) into yeast in duplicate. The paired plasmid libraries were transformed into the yeast strain BY4742 (*MATα ura3Δ0 leu2Δ0 his3Δ1 lys2Δ0*; strain name MSY2; *Brachmann et al., 1998*) using a standard large-scale high efficiency lithium acetate transformation protocol (*Gietz and Schiestl, 2007*) and plated onto 15 cm plates with 2% dextrose, using 1:1000 and 1:10,000 dilutions on 10 cm plates to estimate colony counts, which produced approximately 100,000 colonies. Colonies were washed off plates with 25 mL media with 2% dextrose.

To start the yeast growth assay, six cultures were seeded at low density ($OD_{600}$ measurement of 0.01) in 40 mL media with 2% dextrose and grown overnight for 16 hr. The following morning cultures were moved from 30°C to 4°C to pause growth, then restarted at an $OD_{600}$ of 0.25 in 40 mL media with 2% dextrose 3 hr prior to GAL induction. After 3 hr, once all cultures were in a log growth phase, approximately $1×10^8$ cells were taken from each culture as the starting timepoint sample (designated as '0 hours'). All timepoint samples were spun down in a 1.5 mL Eppendorf tube at 5000 rpm for 1 min, after which supernatant was removed and the pellet was frozen at –80 °C. Approximately $2×10^8$ cells from the remaining cultures were pelleted in 50 mL conical tubes at 3000 rpm, supernatant was removed, and cells were resuspended in 80 mL CSM-Ura media with 2% galactose ($OD_{600}$ measurement of 0.125 $OD_{600}$) to induce PKR expression. Cultures were incubated overnight at 30 °C, 200 rpm. Additional samples were harvested at 12, 16, and 20 hr post-PKR induction, with approximately $1×10^8$ cells taken at each timepoint. The culture density was monitored and back diluted to maintain a log growth phase ($OD_{600}$ measurement less than 1). With three K3 allele conditions (K3, K3Δ58, and K3-H47R), four timepoints (0, 12, 16, and 20 hr) and two replicates, a total of 24 samples were taken across the yeast growth assay.

## Plasmid extraction and barcode amplification

To quantify changes in barcode abundance between timepoints in the yeast growth assay, we harvested plasmids from the four timepoints for each K3 allele, then amplified and sequenced the adjacent PKR and K3 barcodes in each plasmid. Plasmids were harvested from the yeast samples using a modified QIAprep spin miniprep protocol (*Singh and Weil, 2002*). Sampled cell pellets were first thawed at room temperature, followed by the addition of 250 µL QIAprep P1 buffer and 2 µL zymolyase (2.5 units per µL), and incubation at 37 °C for 30 min, followed by adding 250 µL P2 buffer and following the manufacturer instructions for the remainder of the protocol.

To identify and quantify the abundance of each PKR variant paired with each of the three K3 alleles, we performed Illumina sequencing of the PKR barcodes from the plasmids. We used a two-step PCR protocol, the first to amplify the PKR barcode from the plasmids and the second to attach Illumina adapters and Nextera indices for downstream sequencing and sample demultiplexing. For the first PCR reaction, we designed forward and reverse primers as a pool of five primers each, that had 0–4 'N' bases between the Nextera transposase adapter sequence and the priming sequence to stagger the base signal per cycle and maintain sequence diversity across the flow cell (*Wu et al., 2015*). The forward primers (Oligos 44–48) with a melting temperature of 60 °C were pooled together and used for all PCR-1 reactions, annealing immediately upstream of the PKR barcode locus. Five reverse primers were designed and pooled for amplification from each of the three K3 alleles that annealed within the three distinct K3 barcode sequences (Oligos 49–53 for K3, Oligos 54–58 for K3Δ58, and Oligos 59–63 for K3-H47R). One microliter of each primer pool was combined with 50 ng of sample

plasmid DNA and 25 µL Kapa Hifi Hotstart ReadyMix (Kapa Biosystems Cat# KK2602), and topped off to a total volume of 50 µL with water. PCR cycling was performed as follows: (1) 95 °C for 3 min, (2) 98 °C for 20 s, (3) 65 °C for 15 s, (4) 72 °C for 1 min, repeat steps 2–4 for a total of 18 cycles, (5) 72 °C for 1 min, (6) 12 °C hold. PCR products were purified using a MinElute PCR purification kit (QIAGEN Cat#28006) using 11 µL water for the final elution, then quantified using a Qubit dsDNA High Sensitivity (HS) kit (Thermo Fisher Scientific Cat#Q32851).

For the second PCR reaction, 10 ng of amplicon DNA from the first PCR reaction was combined with 10 µL of Nextera adapter index primers (Illumina Cat#20027213) and 25 µL Kapa HiFi Hotstart ReadyMix, and topped off to a total volume of 50 µL with water. PCR cycling was performed as follows: (1) 100 °C for 45 s, (2) 100 °C for 15 s, (3) 60 °C for 30 s, (4) 72 °C for 30 s, repeat steps 2–4 for a total of 8 cycles, (5) 72 °C for 1 min, (6) 12 °C hold. PCR products were purified using a MinElute PCR purification kit using 11 µL of water for the final elution, then quantified using a Qubit dsDNA Broad Range (BR) kit (Thermo Fisher Scientific Cat#Q32850). 500 ng of the second PCR reaction amplicons were gel purified on a Size Select II E-Gel (Thermo Fisher Scientific Cat#G661012) for approximately 13 min to extract the approximately 250 bp amplicon band, followed by quantification with a Qubit dsDNA HS kit. Amplicon samples were diluted to 4 nM before being pooled together, followed by manufacture denature and dilution protocols (Illumina Document#15039740 v10) before sequencing on an Illumina NextSeq 2000 instrument.

## PKR functional scores and screening analysis

Next, we extracted PKR barcode sequences from the Illumina reads and mapped the barcodes back to their corresponding select nonsynonymous variants using the table of PKR barcodes paired with genetic variants generated from the PacBio CCS HiFi reads. Paired reads were assembled into contiguous sequences using PEAR v0.9.11 (*Zhang et al., 2014*), followed by Bartender v1.1 (*Zhao et al., 2018*) to extract and cluster PKR barcodes using the barcode search pattern 'CAAGG[25-27]GGTGA'.

We wrote Python v3.8.18 scripts to tally the PKR barcodes and map them back to genetic variants using the table generated from the PacBio CCS HiFi reads. Barcode counts were normalized to the total read count for each of the four timepoints across each of the K3 alleles, followed by a fold change calculation for each PKR barcode across timepoints 0, 12, 16, and 20 hr:

$$FC_{TP} = -log_2 \left( \frac{Normalized\ Reads_{TP}}{Normalized\ Reads_0} \right)$$

where $FC_{TP}$ is the fold change in normalized reads for a PKR barcode from 0 hr to a subsequent timepoint (TP), *Normalized Reads$_{TP}$* is the read count for a given barcode divided by the total number of reads at the given timepoint, and *Normalized Reads$_0$* is the read count for a given barcode divided by the total number of reads at the 0 hr timepoint. Of note, the log$_2$ of the fold change is inverted such that functional PKR variants that inhibit yeast growth and decreased in abundance were assigned positive fold change value. We then calculated PKR functional scores for each PKR barcode in each of the K3 alleles by calculating the area under the curve using the composite trapezoidal rule. We calculated the PKR functional score for each variant for each K3 allele by averaging the PKR functional score across all representative barcodes for a given variant. As we found a strong correlation between replicates (Pearson correlation coefficient >0.98 for each K3 allele, *Figure 1—figure supplement 3*), we proceeded by combining all reads from the two replicate experiments for each K3 allele and recalculating the fold changes and PKR functional scores for each barcode as described above.

## Predicted PKR complexes and substrate contacts

All molecular graphics and analyses were performed using USCF ChimeraX v1.5 and PyMol v2.5.4. To define PKR residues contacting K3 and eIF2α we used the AlphaFold2 structure prediction tool (ColabFold v1.5.5) in UCSF ChimeraX v1.5 (*Meng et al., 2023*; *Mirdita et al., 2022*; *Figure 1—figure supplement 7*; *Supplementary file 6*, *Supplementary file 7*). We aligned these predictions to existing crystal structures (PDB 2A1A and 1LUZ) and found the AlphaFold2 predictions to largely represent the crystal structures RMSD <1 angstrom for both alignments (*Figure 1—figure supplement 7*; *Supplementary file 5*; *Dar et al., 2005*; *Dar and Sicheri, 2002*). We note the low confidence score (pLDDT <50) of the eIF2α flexible loop containing Ser51, which adopts a compact helical conformation in the predicted model and closely matches the conformation observed in the crystal structure

of unbound yeast eIF2α (PDB 1Q46) (*Dhaliwal and Hoffman, 2003*). As Ser51 of eIF2α would need to travel approximately 17 Å to reach PKR's active site (*Dar et al., 2005*), this predicted structure is likely reflective of a pre-phosphorylation complex, and thus would not capture contacts between PKR and eIF2α made during phosphorylation. The corresponding loop is unresolved in the crystal structure of human PKR in complex with eIF2α (PDB 2A1A). Note that we opted not to use the existing structure for defining contact residues, both for consistency with the K3 contact definitions, as there is no PKR-K3 co-crystal, as well as for the potential to capture interactions mediated by residues unresolved in the crystal structure (PDB 2A1A, PKR Asp338-Asn350 and eIF2α Glu49-Lys60).

To identify potential PKR contact sites from the predicted complex structures, we used PyMol v2.5.4 to select all PKR residue branches within 5 Å of K3 or eIF2α using the command: 'sele contacts, br. /{pdb file name}//A within 5 of /{pdb file name}//B' with chain 'A' being PKR and chain 'B' being either K3 or eIF2α. Predicted PKR sites that contact K3 were: 275, 276, 278, 304, 339, 343, 345, 375, 379, 382, 414, 416, 435, 448, 449, 450, 451, 452, 453, 455, 460, 485, 486, 487, 488, 489, 490, 492, 493, and 496. Predicted PKR sites that contact eIF2α were: 274, 275, 276, 279, 335, 337, 338, 339, 340, 341, 342, 379, 382, 451, 452, 453, 483, 486, 487, 488, 489, 490, 491, 492, and 493.

PKR sites in the kinase domain under positive selection across vertebrate species were identified by *Rothenburg et al., 2009*: 261, 269–272, 307, 314, 322, 360, 368, 375, 378, 379, 382, 385, 389, 394, 405, 428, 448, 449, 462, 471, 483, 486, 488, 491, 493, 500, 502, 504, 505, 514, 520, and 524. Sites conserved across vertebrate PKR homologs were also identified by *Rothenburg et al., 2009* which were used for supplementary analysis: 263, 267, 276–279, 281, 283, 296, 298, 308, 309, 312, 315, 317, 319, 320, 323, 327, 362, 364–367, 369, 374, 377, 397, 401, 404, 406, 407, 410–417, 419, 420, 429–433, 437, 444–446, 450, 451, 454, 455, 457–459, 465, 469, 470, 474–477, 480, 490, 511, 519, 523, and 526.

## Acknowledgements

We thank Thomas Dever, Stefan Rothenburg, Adam Phillippy, Bernard Moss, Stephanie Jaquet, Darach Miller, and members of the Sadhu lab for helpful discussions. We thank Thomas Dever and Mark Rose for strains and plasmids. Next-generation sequencing was performed by both the NIH Intramural Sequencing Center (NISC) and the Microarrays and Single-Cell Genomics Core of the National Human Genome Research Institute. This work utilized the computational resources provided by the NIH HPC Biowulf Cluster (http://hpc.nih.gov). This work was supported by the Intramural Research Program of the National Human Genome Research Institute, NIH (1ZIAHG200401).

## Additional information

### Funding

| Funder | Grant reference number | Author |
| --- | --- | --- |
| National Human Genome Research Institute | 1ZIAHG200401 | Michael James Chambers Sophia B Scobell Meru J Sadhu |

The funders had no role in study design, data collection and interpretation, or the decision to submit the work for publication.

### Author contributions

Michael James Chambers, Conceptualization, Formal analysis, Investigation, Writing – original draft, Writing – review and editing; Sophia B Scobell, Formal analysis, Investigation; Meru J Sadhu, Conceptualization, Formal analysis, Writing – original draft, Writing – review and editing

### Author ORCIDs

Michael James Chambers https://orcid.org/0000-0002-0658-6984
Meru J Sadhu https://orcid.org/0000-0002-8636-9102

Reviewer #1 (Public review): https://doi.org/10.7554/eLife.99575.3.sa1

Reviewer #2 (Public review): https://doi.org/10.7554/eLife.99575.3.sa2
Author response https://doi.org/10.7554/eLife.99575.3.sa3

## Additional files

### Supplementary files

- Supplementary file 1. Oligonucleotides used in the study.
- Supplementary file 2. Expanded list of oligonucleotides used to generate PKR variants.
- Supplementary file 3. Plasmids used in this study.
- Supplementary file 4. Table of PKR functional scores used in *Figures 2–5*.
- Supplementary file 5. Root mean square deviation between AlphaFold2-predicted complexes and experimentally determined structures.
- Supplementary file 6. PDB-formatted atomic coordinate file for the AlphaFold2-predicted structure of the kinase domain of human PKR bound to human eIF2α.
- Supplementary file 7. PDB-formatted atomic coordinate file for the AlphaFold2-predicted structure of the kinase domain of human PKR bound to vaccinia K3.
- MDAR checklist

### Data availability

All data generated or analyzed during this study are included in the manuscript and supporting files or are available at https://doi.org/10.5281/zenodo.11095101.

The following dataset was generated:

| Author(s) | Year | Dataset title | Dataset URL | Database and Identifier |
| --- | --- | --- | --- | --- |
| Chambers MJ | 2024 | greenkidneybean/ dms_human_pkr: Deep Mutational Scan of Human PKR Against Vaccinia K3 | https://doi.org/ 10.5281/zenodo. 13788100 | Zenodo, 10.5281/ zenodo.13788100 |

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
