## [Editor Report · eLife Assessment]

This **important** revised report describes the control of the activity of the RNA-activated protein kinase, PKR, by the Vaccinia virus K3 protein. A strength of the manuscript is the powerful combination of a classic yeast-based assay with high-throughput sequencing and its **convincing** experimental use to characterize large numbers of PKR variants, now with improved controls for potential biases. A minor current limitation that the authors may address in the future is the scope of the screen in terms of the segments of PKR included.

---

## [Referee Report · Reviewer #1 (Public review)]

Summary:

The report examines the control of the antiviral RNA-activated protein kinase, PKR, by the Vaccinia virus K3 protein. K3 binds to PKR, hindering its ability to control protein translation by blocking its phosphorylation of the eukaryotic initiation factor EIF2α. Kinase function is probed by saturation mutation of the K3/EIF2α-binding surface on PKR, guided by models of their interaction. The findings identify specific residues at the predicted interface that asymmetrically influence repression by K3 and the phosphorylation of EIF2α. This recognises the potential of PKR alleles to resist control by the viral virulence factor.

Strengths:

The experimentation is diligent, generating and screening many point mutants to identify residues at the interface between PKR and EIF2α or K3 that distinguishes PKR's phosphor control of its substrate from the antithetical interaction with the viral virulence factor.

Weaknesses:

The protein interaction between PKR and K3 has already been well-explored through phylogenetic and functional analyses and molecular dynamics studies, as well as with more limited site-directed mutational studies using the same experimental assays. Accordingly, the findings are not pioneering but reinforce and extend what had previously been established.

The authors responded to this comment by pointing out that their more comprehensive screen better defined the extent of the plasticity of the K3/EIF2α-binding surface on PKR.

Also in their response, the authors added the caveat that the equivalent expression of the different PKR mutants has not been verified, added information clarifying the states of the model proteins compared to their determined molecular structures, and provided clarifications or responses to all other questions.

I question eLife's assessment that the development of the yeast-based assay is a key advancement of this report, as this assay has been used for over 30 years.

---

## [Referee Report · Reviewer #2 (Public review)]

Chambers et al. (2024) present a systematic and unbiased approach to explore the evolutionary potential of the kinase domain of the human antiviral protein kinase R (PKR) to evade inhibition by a poxviral antagonist while maintaining one of its essential functions.

The authors generated a library of 426 single-nucleotide polymorphism (SNP)-accessible non-synonymous variants of PKR kinase domain and used a yeast-based heterologous virus-host system to assess PKR variants' ability to escape antagonism by the vaccinia virus pseudo-substrate inhibitor K3. The study identified determinant sites in the PKR kinase domain that harbor K3-resistant variants, as well as sites where variation leads to PKR loss of function. The authors found that multiple K3-resistant variants are readily available throughout the domain interface and are enriched at sites under positive selection. They further found some evidence of PKR resilience to viral antagonist diversification. These findings highlight the remarkable adaptability of PKR in response to viral antagonism by mimicry.

Significance of the findings: The findings are important with implications to various fields, including evolutionary biology, virus-host interfaces, genetic conflicts, antiviral immunity.

Strength of the evidence: Convincing methodology using state-of-the-art mutational scanning approach in an elegant and simple setup to address important challenges in virus-host molecular conflicts and protein adaptations.

Strengths

Systematic and Unbiased Approach: The study's comprehensive approach to generating and characterizing a large library of PKR variants provides valuable insights into the evolutionary landscape of PKR kinase domain. By focusing on SNP-accessible variants, the authors ensure the relevance of their findings to naturally occurring mutations.

Identification of Key Sites: The identification of specific sites in the PKR kinase domain that confer resistance or susceptibility to a poxvirus pseudosubstrate inhibition is a significant contribution.

Evolutionary Implications: The authors performed meticulous comparative analyses throughout the study between the functional variants from their mutagenesis screen ("prospective") and the evolutionarily-relevant past adaptations ("retrospective").

Experimental Design: The use of a yeast-based assay to simultaneously assess PKR capacity to induce cell growth arrest and susceptibility/resistance to various VACV K3 alleles is an efficient approach. The combination of this assay with high-throughput sequencing allows for the rapid characterization of a large number of PKR variants.

Areas of improvement

Validation of the screen: In the revised version, the authors now provide the results of two independent experiments in a complete yeast growth assay on a handful of candidates to control the screen's results. This strengthens the direct findings from the screen. It would strengthen the study to complement this validation by another method to assess PKR functions; for example, in human PKR-KO cells, because results between yeast and human cells can differ. These limitations are now acknowledged in the revised version.

Evolutionary Data: Beyond residues under positive selection, the screen allows the authors to also perform a comparative analysis with PKR residues under purifying selection. Because they are assessing one of the most conserved ancestral functions of PKR (i.e. cell translation arrest), it may also be of interest to discuss these highly conserved sites. The authors now discuss the implications for the conserved residues.

Mechanistic insights and viral diversity: While the study identifies key sites and residues involved in vaccinia K3 resistance, it could benefit from further investigation into the underlying molecular mechanisms and the diversity of viral antagonists. The authors have now acknowledged these limitations in the Discussion and updated the manuscript to be more specific. These exciting research avenues will be the objectives of a next study.

Overall Assessment

The systematic approach, identification of key sites, and evolutionary implications are all notable strengths. While there is room for a stronger validation of the functions and further investigation into the mechanistic details and broader viral diversity, the findings are robust and already provide important advancements. The manuscript is well-written and clear, and the revised figures are informative and improved.

---

## [Author Response]

Author response:

The following is the authors’ response to the original reviews.

**Reviewer #1 (Public Review):**
Summary:The report describes the control of the activity of the RNA-activated protein kinase, PKR, by the Vaccinia virus K3 protein. Repressive binding of K3 to the kinase prevents phosphorylation of its recognised substrate, EIF2α (the α subunit of the Eukaryotic Initiation Factor 2). The interaction of K3 is probed by saturation mutation within four regions of PKR chosen by modelling the molecules' interaction. They identify K3-resistant PKR variants that recognise that the K3/EIF2α-binding surface of the kinase is malleable. This is reasonably interpreted as indicating the potential adaptability of this antiviral protein to combat viral virulence factors.Strengths:This is a well-conducted study that probes the versatility of the antiviral response to escape a viral inhibitor. The experimentation is very diligent, generating and screening a large number of variants to recognise the malleability of residues at the interface between PKR and K3.Weaknesses:(1) These are minor. The protein interaction between PKR and K3 has been previously well-explored through phylogenetic and functional analyses and molecular dynamics studies, as well as with more limited site-directed mutational studies using the same experimental assays.Accordingly, these findings largely reinforce what had been established rather than making major discoveries.

First, thank you for your thoughtful feedback. We agree that our results are concordant with previous findings and recognize the importance of emphasizing what we find novel in our results. We have revised the introduction (lines 65-74 of the revised_manuscript.pdf) to emphasize three findings of interest: (1) the PKR kinase domain is largely pliable across its substrate-binding interface, a remarkable quality that is most fully revealed through a comprehensive screen, (2) we were able to differentiate variants that render PKR nonfunctional from those that are susceptible to Vaccinia K3, and (3) we observe a strong correlation between PKR variants that are resistant to K3 WT and K3-H47R.

There are some presumptions:(2) It isn't established that the different PKR constructs are expressed equivalently so there is the contingency that this could account for some of the functional differences.

This is an excellent point. We have revised the manuscript to raise this caveat in the discussion (lines 247-251). One indirect reason to suppose that expression differences among our PKR variants are not a dominant source of variation is that we did not observe much variation in kinase activity in the absence of K3.

(3) Details about the confirmation of PKR used to model the interaction aren't given so it isn't clear how accurately the model captures the active kinase state. This is important for the interaction with K3/EIF2α.

We have expanded on Supplemental Figure 12 and our description of the AlphaFold2 models in the Materials and Methods section (lines 573-590). We clarify that these models may not accurately capture the phosphoacceptor loop of eIF2α (residues Glu49-Lys60) and the PKR β4-5 linker (Asp338-Asn350) as these are highly flexible regions that are absent in the existing crystal structure complex (PDB 2A1A) and have low AlphaFold2 confidence scores (pLDDT < 50). We also noted, in the Materials and Methods section and in the caption of Figure 1, that the modeled eIF2α closely resembles the crystal structure of standalone yeast eIF2α, which places the Ser51 phosphoacceptor site far from the PKR active site. Thus, we expect there are additional undetermined PKR residues that contact eIF2α.

(4) Not all regions identified to form the interface between PKR and K3 were assessed in the experimentation. It isn't clear why residues between positions 332-358 weren't examined, particularly as this would have made this report more complete than preceding studies of this protein interaction.

Great questions. We designed and generated the PKR variant library based on the vaccinia K3 crystal structure (PDB 1LUZ) aligned to eIF2α in complex with PKR (PDB 2A1A), in which PKR residues 338-350 are absent. After the genesis of the project, we generated the AlphaFold2-predicted complex of PKR and vaccinia K3, and have become very interested in the β4-β5 linker, a highly diverse region across PKR homologs which includes residues 332-358. However, this region remains unexamined in this manuscript.

**Reviewer #2 (Public Review):**
Chambers et al. (2024) present a systematic and unbiased approach to explore the evolutionary potential of the human antiviral protein kinase R (PKR) to evade inhibition by a poxviral antagonist while maintaining one of its essential functions.The authors generated a library of 426 single-nucleotide polymorphism (SNP)-accessible non-synonymous variants of PKR kinase domain and used a yeast-based heterologous virus-host system to assess PKR variants' ability to escape antagonism by the vaccinia virus pseudo-substrate inhibitor K3. The study identified determinant sites in the PKR kinase domain that harbor K3-resistant variants, as well as sites where variation leads to PKR loss of function. The authors found that multiple K3-resistant variants are readily available throughout the domain interface and are enriched at sites under positive selection. They further found some evidence of PKR resilience to viral antagonist diversification. These findings highlight the remarkable adaptability of PKR in response to viral antagonism by mimicry.Significance of the findings:The findings are important with implications for various fields, including evolutionary biology, virus-host interfaces, genetic conflicts, and antiviral immunity.Strength of the evidence:Convincing methodology using state-of-the-art mutational scanning approach in an elegant and simple setup to address important challenges in virus-host molecular conflicts and protein adaptations.Strengths:Systematic and Unbiased Approach:The study's comprehensive approach to generating and characterizing a large library of PKR variants provides valuable insights into the evolutionary landscape of the PKR kinase domain. By focusing on SNP-accessible variants, the authors ensure the relevance of their findings to naturally occurring mutations.Identification of Key Sites:The identification of specific sites in the PKR kinase domain that confer resistance or susceptibility to a poxvirus pseudosubstrate inhibition is a significant contribution.Evolutionary Implications:The authors performed meticulous comparative analyses throughout the study between the functional variants from their mutagenesis screen ("prospective") and the evolutionarily-relevant past adaptations ("retrospective").Experimental Design:The use of a yeast-based assay to simultaneously assess PKR capacity to induce cell growth arrest and susceptibility/resistance to various VACV K3 alleles is an efficient approach. The combination of this assay with high-throughput sequencing allows for the rapid characterization of a large number of PKR variants.Areas for Improvement:(5) Validation of the screen: The results would be strengthened by validating results from the screen on a handful of candidate PKR variants, either using a similar yeast heterologous assay, or - even more powerfully - in another experimental system assaying for similar function (cell translation arrest) or protein-protein interaction.

Thank you for your thoughtful feedback. We agree that additional data to validate our findings would strengthen the manuscript. We have individually screened a handful of PKR variants in duplicate using serial dilution to measure yeast growth, and found that the results generally support our original findings. We have revised the manuscript to include these validation experiments (lines 117-119 of the revised_manuscript.pdf, Supplemental Figure 4).

(6) Evolutionary Data: Beyond residues under positive selection, the screen would allow the authors to also perform a comparative analysis with PKR residues under purifying selection. Because they are assessing one of the most conserved ancestral functions of PKR (i.e. cell translation arrest), it may also be of interest to discuss these highly conserved sites.

This is a great point. We do find that there are regions of the PKR kinase domain that are not amenable to genetic perturbation, namely in the glycine rich loop and active site. We contrast the PKR functional scores at conserved residues under purifying selection with those under positive selection in Figure 2E (lines 141-143).

(7) Mechanistic Insights: While the study identifies key sites and residues involved in vaccinia K3 resistance, it could benefit from further investigation into the underlying molecular mechanisms. The study's reliance on a single experimental approach, deep mutational scanning, may introduce biases and limit the scope of the findings. The authors may acknowledge these limitations in the Discussion.

We agree that further investigation into the underlying molecular mechanisms is warranted and we have revised the manuscript to acknowledge this point in the discussion (lines 284-288).

(8) Viral Diversity: The study focuses on the viral inhibitor K3 from vaccinia. Expanding the analysis to include other viral inhibitors, or exploring the effects of PKR variants on a range of viruses would strengthen and expand the study's conclusions. Would the identified VACV K3-resistant variants also be effective against other viral inhibitors (from pox or other viruses)? or in the context of infection with different viruses? Without such evidence, the authors may check the manuscript is specific about the conclusions.

This is a fantastic question that we are interested in exploring in our future studies. In the manuscript we note a strong correlation between PKR variants that evade vaccinia wild-type K3 and the K3-H47R enhanced allele, but we are curious to know if this holds when tested against other K3 orthologs such as variola virus C3. That said, we have revised the manuscript to clarify this limitation to our findings and specify vaccinia K3 where appropriate.

**Reviewer #3 (Public Review):**
Summary:- This study investigated how genetic variation in the human protein PKR can enable sensitivity or resistance to a viral inhibitor from the vaccinia virus called K3.- The authors generated a collection of PKR mutants and characterized their activity in a high-throughput yeast assay to identify (1) which mutations alter PKR's intrinsic biochemical activity, (2) which mutations allow for PKR to escape from viral K3, and (3) which mutations allow for escape from a mutant version of K3 that was previously known to inhibit PKR more efficiently.- As a result of this work, the authors generated a detailed map of residues at the PKR-K3 binding surface and the functional impacts of single mutation changes at these sites.Strengths:- Experiments assessed each PKR variant against three different alleles of the K3 antagonist, allowing for a combinatorial view of how each PKR mutant performs in different settings.- Nice development of a useful, high-throughput yeast assay to assess PKR activity, with highly detailed methods to facilitate open science and reproducibility.- The authors generated a very clean, high-quality, and well-replicated dataset.Weaknesses:(9) The authors chose to focus solely on testing residues in or near the PKR-K3 predicted binding interface. As a result, there was only a moderately complex library of PKR mutants tested. The residues selected for investigation were logical, but this limited the potential for observing allosteric interactions or other less-expected results.

First, we greatly appreciate all your feedback on the manuscript, as well as raising this particular point. We agree that this is a moderately complex library of PKR variants, from which we begin to uncover a highly pliable domain with a few specific sites that cannot be altered. We have revised the manuscript to raise this limitation (lines 284-288 of the revised_manuscript.pdf) and encourage additional exploration of the PKR kinase domain.

(10) For residues of interest, some kind of independent validation assay would have been useful to demonstrate that this yeast fitness-based assay is a reliable and quantitative readout of PKR activity.

We agree that additional data to validate our findings would strengthen the manuscript. We have individually screened a handful of PKR variants in duplicate using serial dilution to measure yeast growth, and generally found that the results support our original findings. We have revised the manuscript to include this validation experiment (lines 117-119, Supplemental Figure 4).

(11) As written, the current version of the manuscript could use more context to help a general reader understand (1) what was previously known about these PKR and K3 variants, (2) what was known about how other genes involved in arms races evolve, or (3) what predictions or goals the authors had at the beginning of their experiment. As a result, this paper mostly provides a detailed catalog of variants and their effects. This will be a useful reference for those carrying out detailed, biochemical studies of PKR or K3, but any broader lessons are limited.

Thank you for bringing this to our attention. We have revised the introduction of the manuscript to provide more context regarding previous work demonstrating an evolutionary arms race between PKR and K3 and how single residue changes alter K3 resistance (lines 51-64).

(12) I felt there was a missed opportunity to connect the study's findings to outside evolutionary genetic information, beyond asking if there was overlap with PKR sites that a single previous study had identified as positively selected. For example, are there any signals of balancing selection for PKR? How much allelic diversity is there within humans, and are people typically heterozygous for PKR variants? Relatedly, although PKR variants were tested in isolation here, would the authors expect their functional impacts to be recessive or dominant, and would this alter their interpretations? On the viral diversity side, how much variation is there among K3 sequences? Is there an elevated evolutionary rate, for example, in K3 at residues that contact PKR sites that can confer resistance? None of these additions are essential, but some kind of discussion or analysis like this would help to connect the yeast-based PKR phenotypic assay presented here back to the real-world context for these genes.

We appreciate this suggestion to extend our findings to a broader evolutionary context. There is little allelic diversity of PKR in humans, with all nonsynonymous variation listed in gnomAD being rare. (PKR shows sequence diversity in comparisons across species, including across primates.) Thus, barring the possibility of variation being present in under-studied populations, there is unlikely to be balancing selection on PKR in humans. Our expectation is that beneficial mutations in PKR for evading a pseudosubstrate inhibitor would be dominant, as a small amount of eIF2α phosphorylation is capable of halting translation (Siekierka, PNAS, 1984). There is a recent report citing PKR missense variants associated with dystonia that can be dominantly or recessively inherited (Eemy et al. 2020 PMID 33236446). Elde et al. 2009 (PMID 19043403) notes that poxvirus K3 homologs are under positive selection but no specific residues have been cited to be under positive selection. The lack of allelic diversity in PKR in humans notwithstanding, PKR could experience future selection in the human population as evidenced by its rapid evolution in primates, so we fully agree that a connection to the real-world context is useful. We have noted these topics in the discussion section (lines 289-294).

**Reviewer #1 (Recommendations For The Authors):**
I have no major criticisms but ask for some clarifications and make some comments about the perceived weaknesses.(13) If the authors disagree with my summation that the findings largely replicate what was known, could they detail how the findings differ from what was known about this protein interaction and the major new insights stemming from the study? Currently, the abstract is a little philosophical rather than listing the explicit discoveries of the study.

Thank you again for raising the need for us to clearly convey the novelty of our findings. We have revised the final paragraph in our introduction as described in comment #1.

(14) As the experimental approach is well reported it is unnecessary to confirm the proposed activity by, for instance, measures of Sui2 phosphorylation. However, previous reports have recognised that point mutants of PKR can be differentially expressed. The impact of this potential effect is unknown in the current experimentation as there are no measures of the expression of the different mutant PKR constructs. The large number of constructs used makes this verification onerous. The potential impact could be ameliorated by redundant replacing each residue (hoping different residues have different effects on expression). Still, this limitation of the study should be acknowledged in the text.

We greatly appreciate this comment and agree that this should be made clear in the text, which we have added to the discussion of the manuscript (lines 247-251).

(15) Preceding findings and the modeling in this report recognise an involvement in the kinase insert region (residues 332 to 358) in PKR's interaction with K3 but this region is excluded from the analysis. These residues have been largely disregarded in the preceding analysis (it is absent from the molecular structure of the kinase) so its inclusion here might have lent a more novel aspect or delivered a more complete investigation. Is there a justification for excluding this flexible loop?

The PKR variant library was designed based on the crystal structure of K3 (PDB 1LUZ) aligned to eIF2α in complex with PKR (PDB 2A1A). After the library was designed and made we attained complete predicted structures of PKR in complex with eIF2α and K3, which largely agrees with the predicted crystal structures but contain the additional flexible loops that were not captured in the crystal structures. Though the library studied here does not explore variation in the kinase insert region, we are very interested in doing so in our future studies.

(16) Could the explanation of the 'PKR functional score' be clarified? The description given within the legend of SF1 was helpful, so could this be replicated earlier in the main body of the text when introducing these experiments? e.g. As PKR activity is toxic to yeast, the number of cells in the pool expressing the functional PKR will decrease over time. Thus the associated barcode read count will also decrease, while the read count for the nonfunctional PKR will increase. This is termed the PKR function score, which will be relatively lower for cells transformed with less active PKR than those with more active PKR.

Thank you for suggesting this clarification, we have revised the manuscript to clarify our definition of the PKR functional score (lines 106-109).

(17) Another suggestion to clarify this term is to modify the figures. Currently, the intent of the first simulated graph in Fig 1E is clear but the inversion of the response (shown by the transposition of the colours) in the next graph (to the right) is less immediately obvious. Accordingly, the orientation of the 'PKR functional score' is uncertain. Could the authors add text to the rightmost graphic in Figure 1E by, for instance, indicating the PKR activity in the vertical column with text such as 'less active' (at the bottom), 'WT' (in the centre), and 'more activity' (at the top)? Also, the position of the inactive K296R mutant might be added to Figure 2A complementing the positioning of the active WT kinase in the first data graph of this kind.

We appreciate your specific feedback to improve the figures of the manuscript, we have made adjustments to Figure 1E to clarify how we derive the PKR functional scores.

(18) The authors don't use existing structures of PKR in their modelling. However, there is no information about the state of the PKR molecule used for modelling. Specific elements of the kinase domain affect its interaction with K3 so it would be informative to know the orientation of these elements in the model. Could the authors detail the state of pivotal kinase elements in their models? This could involve the alignment of the N- and C-lobes, the orientation of kinase spines (C- and R-spines), and the phosphorylation stasis of residues in the activation loop, or at least the position of this loop in relationship to that adopted in the active dimeric kinase (e.g. PDB-2A1A, 3UIU or 6D3L). Alternatively, crystallographic structures of active inactive PKR could be overlayed with the theoretical structure used for modelling (as supplementary information).

We have revised the manuscript to describe the alignment of the predicted PKR-K3 complex with active and inactive PKR, and we have extended Supplemental Figure 12 with an overlay of the predicted structures with existing structures. We have also added a supplemental data file containing the RMSD values of PKR (from the predicted PKR-K3 complex) aligned to active (PDB 2A1A) and inactive (PDB 3UIU) or unphosphorylated (PDB 6D3L) PKR (5_Structure-Alignment-RMSD-Values.xlsx). We have also provided the AlphaFold2 best model predictions for the PKR-eIF2α complex (6_AF2_PKR-KD_eIF2a.pdb) and PKR-K3 complex (7_AF2_PKR-KD_VACV-K3.pdb). Looking across the RMSD values, the AlphaFold2 model of PKR most closely resembles unphosphorylated PKR (PDB 6D3L) though we note the activation loop is absent from PDB 6D3L and 3UIU. We also aligned the Ser51 phosphoacceptor loop of AlphaFold2 eIF2α model to PDB 1Q46 and we see that the model reflects the pre-phosphorylation state. This loop is expected to interact with the PKR active site, which is not captured in our model and we state this explicitly in the caption of Figure 1 (lines 665-668).

(19) Could some specific residue in Figure 7 be labelled (numbered) to orient the findings? Also, the key in this figure doesn't title the residues coloured white (RE red/black/blue). The white also isn't distinguished from the green (outside the regions targeted for mutagenesis).

Excellent suggestion, we have revised this figure to include labels for the sites to orient the reader and clarify our categorization of PKR residues in the kinase domain.

(20) Regarding the discussion, the authors adopt the convention of describing K3 as a pseudosubstrate. Although I realize it is common to refer to K3 as a pseudosubstrate, it isn't phosphorylated and binds slightly differently to PKR so alternative descriptors, such as 'a competitive binder', would more accurately present the protein's function. Possibly for this reason, the authors declared an expectation that evolution pressures should shift K3 to precisely mimic EIF2α. However, closer molecular mimicry shouldn't be expected for two reasons. The first is a risk of disrupting other interactions, such as the EIF2 complex. Secondly, equivalent binding to PKR would demote K3 to merely a stoichiometric competitor of EIF2α. In this instance, effective inhibition would require very high levels of K3 to compete with equivalent binding by EIF2α. This would be demanding particularly upon induction of PKR during the interferon response. To be an effective inhibitor K3 has to bind more avidly than EIF2α and merely requires a sufficient overlap with the EIF2α interface on PKR to disrupt this alternative association. This interpretation predicts that K3 is under pressure to bind PKR by a different mechanism than EIF2α.

We appreciate your thoughtful point about the usage of the term pseudosubstrate. Ultimately, we’ve decided to continue using the term due to its historical usage in the field. The question of the optimal extent of mimicry in K3 is a fascinating one, and we greatly appreciate your thoughts. We wholly agree that the possibility of K3 having superior PKR binding relative to eIF2α would be preferable to perfect mimicry. In our Ideas and Speculation section, we propose that benefits towards increasing PKR affinity may need to be balanced against potential loss of host range resulting from overfitting to a given host’s PKR. However, the possibility that reduced mimicry could be selected to avoid disruption of eIF2 function had not occurred to us; thank you for pointing it out!

(21) The discussion of the 'positive selection' of sites is also interesting in this context. To what extent has the proposed positive selection been quantified? My understanding is that all of the EIF2α kinases are conserved and so demonstrate lower levels of residue change that might be expected by random mutagenesis i.e. variance is under negative selection. The relatively higher rate of variance in PKR orthologs compared to other EIF2α kinases could reflect some relaxation of these constraints, rather than positive selection. Greater tolerance of change may stem from PKR 's more sporadic function in the immune response (infrequent and intermittent presence of its activating stimuli) rather than the ceaseless control of homeostasis by the other EIF2α kinases. Also, induction of PKR during the immune response might compensate for mutations that reduce its activity. I believe that the entire clade of extant poxviruses is young relative to the divergence between their hosts. Accordingly, genetic variance in PKR predates these viruses. Although a change in PKR may become fixed if it affords an advantage during infection, such an advantage to the host would be countered by the much higher mutation rates of the virus. This would appear to diminish the opportunity for a specific mutation to dominate a host population and, thereby, to differentiate host species. Rather, pressure to elude control by a rapidly evolving viral factor would favour variation at sites where K3 binds. This speculation offers an alternative perspective to the current discussion that the variance in PKR orthologs stems from positive selection driven by viral infection.

We appreciate this stimulating feedback for discussion. Three of the four eIF2α kinases (HIR, PERK, and GCN2) appear to be under purifying selection (Elde et al. 2009, PMID 19043403), which stand in contrast to PKR. Residues under positive selection have been found throughout PKR, including the dsRNA binding domains, linker region, and the kinase domain. Importantly, the selection analysis from Elde et al. and Rothenburg et al. concluded that positive selection at these sites is more likely than relaxed selection. We agree that poxviruses are young, though we would guess that viral pseudosubstrate inhibition of PKR is ancient. Many viral proteins have been reported to directly interact with PKR, including herpes virus US11, influenza A virus NS1A, hepatitis C virus NS5A, and human immunodeficiency virus Tat. The PKR kinase domain does contain residues under purifying selection that are conserved among all four eIF2α kinases, but it also contains residues under positive selection that interface with the natural substrate eIF2α. Our work suggests that PKR is genetically pliable across several sites in the kinase domain, and we are curious to know if this pliability would hold at the same sites across the other three eIF2α kinases.

(22) The manuscript is very well written but has a small number of typos; e.g. an aberrant 'e' ln 7 of the introduction, capitalise the R in ranavirus on the last line of the fourth paragraph of the discussion, and eIF2α (EIF2α?) is occasionally written as eIFα in the materials&methods.

Thank you for bringing these typos to our attention! We’ve deleted the aberrant ‘e’ in the introduction, capitalized ‘Ranavirus’ in the discussion (line 265), and corrected ‘eIFα’ to ‘eIF2α’ throughout the manuscript.

**Reviewer #2 (Recommendations For The Authors):**
Additional minor edits or revisions:(23) Paragraph 3 of the Introduction gives the impression that most of the previous work on the PKR-virus arms race is speculative. However, it is one of the best-described and most convincing examples of virus-host arms races. Can the authors edit the paragraph accordingly?

Thank you for bringing this to our attention. We have revised the third paragraph and strengthened the description of the evolutionary arms race between PKR and viral pseudosubstrate antagonists.

(24) Introduction: PKR has "two" double-stranded RNA binding domains. Can the authors update the text accordingly?

We have updated the manuscript to clarify PKR has two dsRNA binding domains (lines 44-45).

(25) The authors test here for one of the key functions of PKR: cell growth/translation arrest. Because of PKR pleiotropy, the manuscript may be edited accordingly: For example, statements such as "We found few genetic variants render the PKR kinase domain nonfunctional" are too speculative as they may retain other (not tested here) functions.

This is a great suggestion, we have revised the manuscript to specify our definition of nonfunction in the context of our experimental screen (lines 86-92 and 106-109) and acknowledge this limitation in our experimental screen (lines 304-307).

(26) The authors should specify "vaccinia" K3 whenever appropriate.

We appreciate this comment and have revised the manuscript to specify vaccinia K3 where appropriate (e.g. lines 62,66, 70, 80, 108, and 226).

(27) Ref for ACE2 diversification may include Frank et al 2022 PMID: 35892217.

Thank you for pointing us to this paper, we have included it as a reference in the manuscript (line 277).

(28) Positive selection of PKR as referred to by the authors corresponds to analyses performed in primates. As shown by several studies, the sites under positive selection may vary according to host orders. Can the authors specify this ("primate") in their manuscript? And/or shortly discuss this aspect.

Thank you for raising this point. In the manuscript we performed our analysis using vertebrate sites under positive selection as identified in Rothenburg et al. 2009 PMID 19043413 (lines 51 and figure legends). We performed the same analysis using sites under positive selection in primates (as identified by Elde et al. 2009 PMID 19043403) and again found a significant difference in PKR functional scores versus K3. We have revised the manuscript to clarify our use of vertebrate sites under positive selection (line 80-81).

(29) We view deep mutational scanning experiments as a complementary approach to positive selection": The authors should edit this and acknowledge previous and similar work of other antiviral factors, in particular one of the first studies of this kind on MxA (Colon-Thillet et al 2019 PMID: 31574080), and TRIM5 (Tenthorey et al 2020 PMID: 32930662).

Thank you for raising up these two papers, which we acknowledge in the revised manuscript (line 299).

(30) We believe Figure S7 brings important results and should be placed in the Main.

We appreciate this suggestion, and have moved the contents of the former supplementary Figure 7 to the main text, in Figure 6.

(31) The title may specify "poxvirus".

Thank you for the suggestion to specify the nature of our experiment, we have adjusted the title to: Systematic genetic characterization of the human PKR kinase domain highlights its functional malleability to escape a poxvirus substrate mimic (line 3).

**Reviewer #3 (Recommendations For The Authors):**
(32) No line numbers or page numbers are provided, which makes it difficult to comment.

We sincerely apologize for this oversight and have included line numbers in our revised manuscript as well as the tracked changes document.

(33) In the introduction, I recommend defining evolutionary arms races more clearly for a broad audience.

Thank you for this suggestion. We have revised the manuscript in the first and third paragraphs to more clearly introduce readers to the concept of an evolutionary arms race.

(34) The introduction could use a clearer statement of the question being considered and the gap in knowledge this paper is trying to address. Currently, the third paragraph includes many facts about PKR and the fourth paragraph jumps straight into the approach and results. Some elaboration here would convey the significance of the study more clearly. As is, the introduction reads a bit like "We wanted to do deep mutational scanning. PKR seemed like an ok protein to look at", rather than conveying a scientific question.

This is a great suggestion to improve the introduction section. We have heavily revised the third and fourth paragraphs of the introduction to clarify the motivation, approach, and significance of our work.

(35) Relatedly, did the authors have any hypotheses at the start of the experiment about what kinds of results they expected? e.g. What parts of PKR would be most likely to generate escape mutants? Would resistant mutants be rare or common? etc? This would help the reader to understand which results are expected vs. surprising.

These are all great questions. We have revised the introduction of the manuscript to point out that previous studies have characterized a handful of PKR variants that evade vaccinia K3, and these variants were made at sites found to be under positive selection (lines 60-64).

(36) A description of the different K3 variants and information about why they were chosen for study should also be added to the Introduction. It was not until Figure 5 that the reader was told that K3-H47R was the same as the 'enhanced' K3 allele you are testing.

Thank you for bringing this to our attention, we have revised the introduction to clarify the experimental conditions (lines 65-67) and specify K3-H47R as the enhanced allele earlier in the manuscript (line 100).

(37) Does every PKR include just a single point mutation? It would be nice to see data about the number and types of mutations in each PRK window added to Supplemental Figure 1.

Thank you for the suggestion to improve this figure. Every PKR variant that we track has a single point mutation that generates a nonsynonymous mutation. In our PacBio sequencing of the PKR variant library we identified a few off-target variants or sequences with multiple variants, but we identified the barcodes linked to those constructs and discarded those variants in our analysis. We have revised Supplemental Figure 1 to include the number and types of mutations made at each PKR window.

(38) In terms of the paper's logical flow, personally, I would expect to begin by testing which variants break PKR's function (Figure 3) and then proceeding to see which variants allow for K3 escape (Figure 2). Consider swapping the order of these sections.

Thank you for this suggestion, and we can appreciate how the flow of the manuscript may be improved by swapping Figures 2 and 3. We have decided to maintain the current order of the figures because we use Figure 3 to emphasize the distinction of PKR sites that are nonfunctional versus susceptible to vaccinia K3.

(39) Figure 3A seems like a less-informative version of Figure 4A, recommend combining these two. Same comment with Figure 5A and Figure 6A.

We appreciate this specific feedback for the figures. Though there are similarities between figure panels (e.g. 3A and 4A) we use them to emphasize different points in each figure. For example, in Figure 3 we emphasize the general lack of variants that impair PKR kinase activity, and in Figure 4 we distinguish kinase-impaired variants from K3-susceptible variants. For this reason, and given space constraints, we have chosen to maintain the figures separately. We did decide to move the former Figure 6 to the supplement.

(40) In general, it felt like there was a lot of repetition/re-graphing of the same data in Figures 3-6. I recommend condensing some of this, and/or moving some of the panels to supplemental figures.

Thank you for your suggestion, we have revised the manuscript and have moved Figure 6 to Supplemental Figure 7.

(41) In contrast, Supplemental Figure 7 is helpful for understanding the distribution of the data. Recommend moving to the main text.

This is a great recommendation, and we have moved Supplemental Figure 7 into Figure 6.

(42) How do the authors interpret an enrichment of positively selected sites in K3-resistant variants, but not K3-H74R-resistant variants? This seems important. Please explain.

Thank you for this suggestion to improve the manuscript; we agree that this observation warranted further exploration. We found a strong correlation in PKR functional scores between K3 WT and K3-H47R, and with that we find sites under positive selection that are resistant to K3 WT are also resistant to K3-H47R. The lack of enrichment at positively selected sites appears to be caused by collapsed dynamic range between PKR wild-type-like and nonfunctional variants in the K3-H47R screen. We have revised the manuscript to clarify this point (line 202-204).

(43) Discussion: The authors compare and contrast between PKR and ACE2, but it would be worth mentioning other examples of genes involved in antiviral arms races wherein flexible, unstructured loops are functionally important and are hotspots of positive selection (e.g. MxA, NLRP1, etc).

We greatly appreciate this suggestion to improve the discussion. We note this contrast between the PKR kinase domain and the flexible linkers of MxA and NLRP1 in the revised manuscript (lines 273-274).

(44) Speculation section: What is the host range of the vaccinia virus? Is it likely to be a generalist amongst many species' PKRs (and if so, how variable are those PKRs)? Would be worth mentioning for context if you want to discuss this topic.

Thank you for raising this question. Vaccinia virus is the most well studied of the poxviruses, having been used as a vaccine to eradicate smallpox, and serves as a model poxvirus. Vaccinia virus has a broad host range, and though the name vaccinia derives from the Latin word “vacca” for cow the viruses origin remains uncertain (Smith 2007 https://doi.org/10.1007/978-3-7643-7557-7_1). has been used to eradicate smallpox as a vaccine and serves as a model poxvirus. Thought the natural host is unknown, it appears to be a general inhibitor of vertebrate PKRs The natural host of vaccinia virus is unknown, though there is some evidence to suggest it may be native to rabbits and does appear to be generalist.

(45) Many papers in this field discuss interactions between PKR and K3L, rather than K3. I understand that this is a gene vs. protein nomenclature issue, but consider matching the K3L literature to make this paper easier to find.

Thank you for bringing this to our attention. We have revised the manuscript to specify that vaccinia K3 is expressed from the *K3L* gene in both the abstract (line 26) and the introduction (line 56) to help make this paper easier to find when searching for “*K3L*” literature.

(46) Which PKR sequence was used as the wild-type background?

This is a great question. We used the predominant allele circulating in the human population represented by Genbank m85294.1:31-1686. We cite this sequence in the Methods (line 421) and have added it to the results section as well (lines 84).

(47) Figure 1C: the black dashed line is difficult to see. Recommend changing the colors in 1A-1C.

Thank you for this suggestion, we have changed the dashed lines from black to white to make them more distinguishable.

(48) Figure 1D: Part of the point of this figure is to convey overlaps between sites under selection, K3 contact sites, and eIF2alpha contact sites, but at this scale, many of the triangles overlap. It is therefore impossible to tell if the same sites are contacted vs. nearby sites. Perhaps the zoomed-in panels showing each of the four windows in the subsequent figures are sufficient?

Thank you for bringing this to our attention. We have scaled the triangles down to reduce their overlap in Figure 1D and list all sites of interest (predicted eIF2α and vaccinia contacts, conserved sites, and positive selection sites) in the Materials and Methods section “Predicted PKR complexes and substrate contacts”.

(49) Figure 1E: under "1,293 Unique Combinations", there is a line between the PKR and K3 variants, which makes it look like they are expressed as a fusion protein. I believe these proteins were expressed from the same plasmid, but not as a fusion, so I recommend re-drawing. Then in the graph, the y-axis says "PKR abundance", but from the figure, it is not clear that this refers to relative abundance in a yeast pool. Perhaps "yeast growth" or similar would be clearer?

Thank you for the specific feedback to improve Figure 1. We have made the suggested edits to clarify that PKR and vaccinia K3 are not fused but each is expressed from their own promoter. We have also changed the y-axis from “PKR Abundance” to “Yeast Growth”.